# One-Step Generalization Ratio Guided Optimization for Domain Generalization

Sumin Cho [* 1]   Dongwon Kim [* 1]   Kwangsu Kim [1]

## Abstract

Domain Generalization (DG) aims to train models that generalize to unseen target domains but often overfit to domain-specific features, known as undesired correlations. Gradient-based DG methods typically guide gradients in a dominant direction but often inadvertently reinforce spurious correlations. Recent work has employed dropout to regularize overconfident parameters, but has not explicitly adjusted gradient alignment or ensured balanced parameter updates. We propose GENIE (Generalization-ENhancing Iterative Equalizer), a novel optimizer that leverages the One-Step Generalization Ratio (OSGR) to quantify each parameter's contribution to loss reduction and assess gradient alignment. By dynamically equalizing OSGR via a preconditioning factor, GENIE prevents a small subset of parameters from dominating optimization, thereby promoting domain-invariant feature learning. Theoretically, GENIE balances convergence contribution and gradient alignment among parameters, achieving higher OSGR while retaining SGD's convergence rate. Empirically, it outperforms existing optimizers and enhances performance when integrated with various DG and single-DG methods.

## 1. Introduction

Deep neural networks (DNNs) achieve high accuracy when training and test data share a similar distribution. However, in real-world applications, data distributions often shift, causing performance degradation(Muandet et al., 2013). Domain Generalization (DG) addresses this issue by training models to generalize to out-of-distribution data from unseen domains. The main challenge is to prevent overfitting

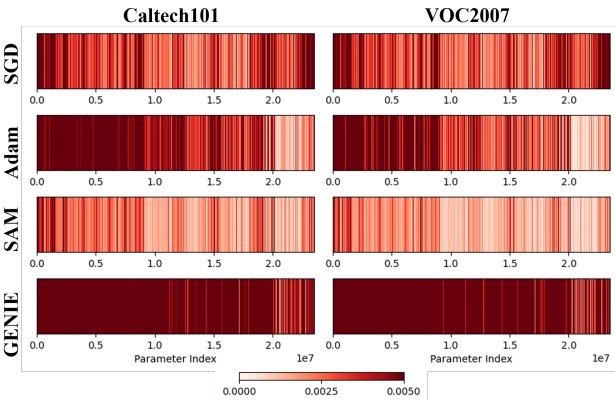

*Figure 1.* Heatmaps visualizing normalized parameter update magnitudes by parameter ID for different optimizers throughout training on the VLCS dataset in the DG. Previous optimizers (SGD, Adam, SAM) exhibit an imbalanced parameter update distribution, where a subset of parameters dominates the optimization process. In contrast, GENIE uniformly adjusts parameter-wise OSGR, mitigating overfitting to specific parameters and promoting a more balanced optimization across the entire parameter space.

to domain-specific features—known as spurious correlations—while learning invariant features and causal relationships that generalize across diverse domains(Shi et al., 2022; Hemati et al., 2023; Shah et al., 2020a; Ye et al., 2024).

Several DG methods have attempted to guide the gradient toward a dominant direction during training (Parascandolo et al., 2021; Shahtalebi et al., 2021; Shi et al., 2022; Rame et al., 2022). However, this dominant direction often itself driven by spurious features, inadvertently reinforcing undesired correlations. This suggests that aligning gradients toward a single dominant direction is insufficient to fully solve the problem, highlighting the need for other perspectives.

A recent approach (Michalkiewicz et al., 2023) introduced a parameter-wise dropout mechanism based on Gradient Signal-to-Noise Ratios (GSNR) to suppress overly predictive parameters and reduce their influence on optimization. While this strategy mitigates parameter updates driven by spurious correlations, it does not adjust the magnitudes of updates based on their individual contributions to general-

---

*Equal contribution [1]Department of Computer Science and Engineering, University of Sungkyunkwan, Suwon, Korea. Correspondence to: Sumin Cho <jsm0707@skku.edu>, Dongwon Kim <kdwaha@skku.edu>, Kwangsu Kim <kim.kwangsu@skku.edu>.

*Proceedings of the 42nd International Conference on Machine Learning*, Vancouver, Canada. PMLR 267, 2025. Copyright 2025 by the author(s).

ization. This raises the open question of how to design optimizers that explicitly balance parameter updates according to their principled contributions to generalization, thereby mitigating the influence of spurious correlations.

Motivated by this perspective, we propose *Generalization-ENhancing Iterative Equalizer (GENIE)*, a novel optimizer for addressing parameter imbalance. Recent work (Liu et al., 2020) introduced the One-Step Generalization Ratio (OSGR) that measures how effectively a single gradient update reduces test loss compared to training loss, providing insight into a model's generalization potential. OSGR reflects the contributions of individual parameters to generalization, based on their convergence speed and degree of gradient alignment. To leverage this insight, GENIE integrates a preconditioning factor that dynamically balances parameter-wise OSGR throughout training. This prevents a small subset of parameters from dominating the optimization, thereby promoting more robust and domain-invariant feature learning.

Our theoretical analysis shows that existing optimizers typically focus on either convergence speed or gradient alignment, often resulting in suboptimal generalization. In contrast, GENIE explicitly balances both, achieving a higher OSGR while maintaining the convergence rate of SGD(Robbins & Monro, 1951) in non-convex settings. We empirically validated GENIE on five standard DG datasets(Li et al., 2017; Fang et al., 2013; Venkateswara et al., 2017; Beery et al., 2018; Peng et al., 2019) where it consistently outperformed established optimizers, even with extended iterations. Furthermore, using our optimizer in existing DG and Single-DG (SDG) algorithms enhances their performance. We summarize our contributions as follows:

- We propose GENIE, a novel optimizer that addresses the overlooked issue of parameter imbalance in DG. It suppresses over-predictive parameters while promoting balanced parameter updates.

- We incorporate OSGR, previously used as a generalization metric, into the optimizer's core principle. This provides an efficient and novel perspective on generalization for addressing DG.

- GENIE is a domain-agnostic optimizer. It is validated across multiple DG benchmarks and SDG tasks, demonstrating its broad applicability and scalability.

## 2. Related Work

### 2.1. Domain Generalization

Existing DG methods address domain shift through two main strategies: (1) Feature Alignment, which aims to align features across domains to ensure consistent optimization,

including methods such as domain-invariant feature learning (Sun & Saenko, 2016; Arjovsky et al., 2019; Krueger et al., 2021), data augmentation (Xu et al., 2020; Yan et al., 2020; Wang et al., 2020), and feature disentanglement (Nam et al., 2021; Mahajan et al., 2021). (2) Gradient Alignment, which focuses on aligning gradients across domains to ensure stable learning dynamics. Representative approaches include minimizing gradient differences (Koyama & Yamaguchi, 2020), increasing gradient inner products (Shi et al., 2022), updating weights only when gradient directions align (Parascandolo et al., 2021; Shahtalebi et al., 2021), and reducing inter-domain gradient variance (Rame et al., 2022). Recently, Sharpness Aware Minima (SAM)(Foret et al., 2021) has improved in-distribution generalization, inspiring the development of optimizers specifically designed for OOD tasks (Zhang et al., 2024; Wang et al., 2023). However, most DG studies overlook imbalanced parameter updates caused by differences in convergence speed or generalization capacity during optimization.

### 2.2. Preconditioning

Preconditioning improves the efficiency of optimization algorithms by incorporating curvature information of the loss function or adjusting the magnitude and direction of parameter updates. It accelerates convergence and enhances stability during training and can be categorized into three main types (Ye, 2024; Amari et al., 2021) (1) Hessian Based Preconditioning: utilizes the inverse or approximations of the Hessian matrix to capture curvature information. (Montavon et al., 2012; Dennis & Moré, 1977) (2) Adaptive Learning Rate Based Preconditioning: dynamically adjusts learning rates based on gradient magnitudes, as seen in optimizers like AdaGrad (Duchi et al., 2011), RMSProp (Hinton et al., 2012), and Adam (Kingma, 2014). (3) Normalization-Based Preconditioning: normalizes inputs and activations, as exemplified by Batch Normalization(Ioffe & Szegedy, 2015), to improve the Hessian's condition number and enhance training stability. Previous preconditioning methods aim to optimize speed and stability. The application of preconditioning to improve model generalization remains underexplored.

## 3. Method

### 3.1. Preliminary

To address the challenge of generalization in unseen target domains, a recent study(Liu et al., 2020) introduced the concept of OSGR $R(Z, n)$. OSGR quantifies how well model updates contribute to generalization by measuring the ratio of loss reduction between test $D'$ and training data $D$ after a single optimization step:

$$R(Z, n) = \frac{\mathbb{E}_{D,D'\sim\mathcal{Z}^n}\Delta L_{D'}}{\mathbb{E}_{D\sim\mathcal{Z}^n}\Delta L_D}, \tag{1}$$

where $\Delta L_{D'}$ and $\Delta L_D$ represent the loss changes on test and training data, respectively. OSGR is influenced by two key factors: (1) the contribution of each parameter to loss reduction, characterized by the gradient magnitude, and (2) the alignment of parameter gradients across the data distribution. Higher OSGR indicates better generalization, reflecting consistent and balanced parameter updates.

To better understand these dynamics, the following theorem links OSGR to parameter-wise statistics:

**Theorem 3.1** (From Paper(Liu et al., 2020)). *The relationship between gradient updates and generalization can be expressed as follows:*

$$R(Z, n) = 1 - \frac{1}{n} \sum_{j \in J} \frac{\mathbb{E}_{D \sim \mathcal{Z}^n}[g_j^2]}{\sum_{j' \in J} \mathbb{E}_{D \sim \mathcal{Z}^n}[g_{j'}^2]} \cdot \frac{1}{r_j + \frac{1}{n}}, \quad (2)$$

*where $J$ denotes the set of parameter index, $g_j^2$ is the squared gradient magnitude, $\rho_j^2$ is the noise variance, and $n$ is the number of samples. Parameters with higher Gradient Signal-to-Noise Ratios (GSNR), defined as $r_j = \frac{g_j^2}{\rho_j^2}$, yield higher OSGR, contributing more significantly to generalization.*

A recent study (Michalkiewicz et al., 2023) leveraged GSNR to suppress overly predictive parameters during training, aiming to prioritize robust features and reduce noisy updates. However, this approach overlooks parameter-wise imbalances in OSGR, which limits overall generalization performance.

In this context, we propose a preconditioning-based approach that dynamically balances OSGR across parameters. By incorporating parameter-specific preconditioning factors, our method ensures that updates are aligned with both gradient magnitude and noise characteristics, preventing overfitting to noisy or well-learned features. This strategy not only enhances generalization but also ensures stable convergence in diverse DG settings.

### 3.2. Proposed Method

Based on Theorem 3.1, Michalkiewicz et al. (2023) introduced a gradient-masking approach that prioritizes updates for parameters with low GSNR, aiming to enhance their contribution to generalization. They argue that boosting updates to low-GSNR parameters can increase the overall GSNR and thus improve the optimization signal-to-gradient ratio (OSGR). Inspired by this perspective, we hypothesize the following relationship:

**Conjecture** *Uniformly distributed OSGR across parameters indicate better generalization performance.*

This conjecture guides the design of our method. Rather than modifying the dropout ratio across parameters, we in-

troduce a preconditioning term that more accurately adjusts the OSGR. Next, we inject noise into all parameters to encourage exploration toward better optima. Finally, we apply random dropout to stabilize parameter updates and reduce overfitting.

#### 3.2.1. PRECONDITIONING

We propose a preconditioning factor $p_j$ to ensure balanced contributions of each parameter to the OSGR, thus enhancing generalization. The key idea is to maintain equitable parameter influence on the overall generalization performance throughout the optimization process. We propose the following corollary for this purpose.

**Corollary 3.2** (Preconditioning and OSGR). *If each parameter $j$ applies a preconditioner $p_j$, the OSGR can be expressed as:*

$$R'(Z, n) = \sum_{j \in J} \frac{p_j \mathbb{E}_{D \sim \mathcal{Z}^n}[g_j^2]}{\sum_{j' \in J} p_{j'} \mathbb{E}_{D \sim \mathcal{Z}^n}[g_{j'}^2]} \cdot \frac{1}{\frac{1}{n \cdot r_j} + 1}, \quad (3)$$

*or equivalently:*

$$R'(Z, n) = 1 - \frac{1}{n} \sum_{j in J} \frac{p_j \mathbb{E}_{D \sim \mathcal{Z}^n}[g_j^2]}{\sum_{j' \in J} p_{j'} \mathbb{E}_{D \sim \mathcal{Z}^n}[g_{j'}^2]} \cdot \frac{1}{r_j + \frac{1}{n}}. \quad (4)$$

From Corollary 3.2, to maintain a balanced influence of parameter $j$ on the overall OSGR, we propose:

$$p_j = \frac{1}{\mathbb{E}_{D \sim \mathcal{Z}^n}[g_j^2]} \left( r_j + \frac{1}{n} \right). \quad (5)$$

This leads to the OSGR:

$$R'(Z, n) = 1 - \frac{1}{n} \sum_{j \in J} \frac{1}{\sum_{j' \in J} \left( r_{j'} + \frac{1}{n} \right)} = 1 - \frac{1}{n \mathbb{E}_{j \in J} \left( r_j + \frac{1}{n} \right)}, \quad (6)$$

where $\mathbb{E}_{j \in J} \left( r_j + \frac{1}{n} \right)$ represents the average GSNR contribution across parameters. Without preconditioning, parameters with large $g_j^2$ but low GSNR may receive higher weights in the OSGR expression, inflating the subtraction term. Our preconditioning alleviates this issue and improves the OSGR. This dynamic adjustment with preconditioning mitigates parameter-wise imbalances, ensuring that well-generalized features are not overwhelmed by noisy or overly dominant parameters.

In implementation, we ignore the $\frac{1}{n}$ term as $n$ is sufficiently large, and clipping variance by $tanh(\frac{1}{\sigma^2})$ for stability. More detailed analysis on influence of variance is described in Section 3.3.3. This preconditioner $p_j$ is straightforward to compute and requires only the gradient statistics $m_t$ and variance $\sigma_t$, which can be estimated during training. This efficiency makes it suitable for a wide range of DG tasks.

### 3.2.2. NOISE INJECTION

To enhance exploration during optimization, we introduce *noise injection*, where a noise term scaled by the variance is added to the gradient. Specifically, the noise scale is determined by $1 - \tanh(\frac{1}{\sigma^2})$, reducing noise for high variance parameters while increasing it for low variance parameters. Motivated by (Mansilla et al., 2021), this injection boosts updates to parameters with low preconditioning value.

### 3.2.3. RANDOM MASK

To further stabilize updates and mitigate overfitting, we apply a *random dropout mask*. This mask, sampled from a Bernoulli distribution, selectively zeroes out gradient components. By applying random masking after the preconditioning step, all parameters are equally considered to ensure robust updates.

### 3.3. Analysis

We provide a comprehensive theoretical analysis of our method from three perspectives. First, we examine generalization through the OSGR, which highlights how our effectively balances OSGR value across parameters. Second, we formalize our approach under the PAC-Bayes framework, showing that our method explicitly minimizes a tighter generalization bound. Finally, we establish that our optimizer retains the convergence rate of standard SGD while enabling more robust generalization. Proofs are provided in Appendix C.

### 3.3.1. GENERALIZATION ANALYSIS WITH OSGR

We obtain the following corollary regarding the OSGR of these optimizers:

**Corollary 3.3** (OSGR of Optimizers). *The OSGR of our proposed optimizer is:*

$$\mathcal{R}_{Ours} = 1 - \frac{1}{n\mathbb{E}_{j \in J}\left(r_j + \frac{1}{n}\right)}, \tag{7}$$

*Comparing the resulting OSGR across different optimizers, we have:*
$$\mathcal{R}_{Ours} \geq \mathcal{R}_{SGD} \approx \mathcal{R}_{Adam}. \tag{8}$$

This corollary demonstrates that our proposed preconditioning achieves better generalization by attaining a higher overall OSGR. The following remarks provide further context and analysis:

*Remark* 3.4 (Conceptual Components of Optimizers). The preconditioning applied by common optimizers can be viewed as the element-wise product of two conceptual components:

- **Convergence Term:** controls the effective step size,

---

**Algorithm 1** Algorithm for GENIE

**Input:** Mini-batches $\{\mathcal{B}_t\}_{t=1}^T$, Learning Rate $\alpha$, Total Steps $T$.
**Hyperparameters:** $\beta \in [0, 1]$, Dropout Probability $p$
**Initialize:** Parameters $\theta_0$, $m_0 \leftarrow 0$, $v_0 \leftarrow 0$.
**for** $t = 1$ to $T$ **do**
    **Compute Gradient:**

$$g_t = \nabla\mathcal{L}(\theta_t; \mathcal{B}_t)$$

    **Update Moving Averages:**

$$m_t \leftarrow \beta m_{t-1} + (1-\beta)g_t, \quad v_t \leftarrow \beta v_{t-1} + (1-\beta)g_t^2$$

    **Calculate GSNR and Preconditioning:**

$$\sigma_t^2 = v_t - m_t^2, \quad r_j = \tanh(\frac{1}{\sigma_t^2})m_t^2$$

$$\hat{g}_t \leftarrow \frac{m_t}{1 - \beta^t} \cdot \frac{1}{v_t} \cdot r_t$$

    **Noise Injection:**

$$Noise_t \leftarrow \xi_t\left[1 - \tanh(\frac{1}{\sigma_t^2})\right], \quad \xi_t \sim \mathcal{N}(0, \sigma^2)$$

    **Random Mask:**

$$M_j \sim Bernoulli(p)$$

$$\hat{g}_t \leftarrow (\hat{g}_t + Noise_t) \odot M$$

    **Update Parameters:**

$$\theta_{t+1} \leftarrow \theta_t - \alpha\tilde{g}_t$$

**end for**
**Output:** Final parameters $\theta_{T+1}$.

---

thus contributing to faster convergence. It includes terms such as $\mathbb{E}_{D\sim\mathcal{Z}^n}[g_j^2]$ or $\mathbb{E}_{D\sim\mathcal{Z}^n}[g_j]$.

- **Alignment Term:** adjusts gradients toward stable directions. It includes the GSNR term $r_j$.

Table 1 summarizes the convergence term, alignment term and their resulting OSGR, including SGD, Adam, and our method.

*Remark* 3.5 (Optimizer-Specific Analysis). SGD maintains a baseline OSGR value with no explicit adjustment. Adam introduces a convergence component combined with a partial alignment factor. In contrast, our method effectively integrates both aspects in a balanced manner.

Overall, this analysis highlights how each optimizer's design affects generalization through gradient alignment and

*Table 1.* Comparison of optimizers with preconditioning split into Convergence and Alignment, and OSGR as a separate column.$\mathbb{E}_W$ means weighted averaging with weight $\sum_j W_j = 1$

| OPT. | PRECONDITIONING | | OSGR | WEIGHT |
|---|---|---|---|---|
| | CONVERGENCE | ALIGNMENT | | |
| SGD | – | – | $1 - \frac{1}{n}\sum_{j \in J} W_j \cdot \frac{1}{r_j + \frac{1}{n}}$ | $W_j = \frac{\mathbb{E}_{D \sim \mathcal{Z}^n}[g_j^2]}{\sum_{j'} \mathbb{E}_{D \sim \mathcal{Z}^n}[g_{j'}^2]}$ |
| ADAM | $\frac{1}{\left|\mathbb{E}_{D \sim \mathcal{Z}^n}(g_j)\right|}$ | $\sqrt{\frac{1}{\frac{1}{n \cdot r_j} + 1}}$ | $\sum_{j \in J} W_j \cdot \frac{1}{\frac{1}{n \cdot r_j} + 1}$ | $W_j = \frac{\sqrt{\mathbb{E}_{D \sim \mathcal{Z}^n}[g_j^2]}}{\sum_{j'} \sqrt{\mathbb{E}_{D \sim \mathcal{Z}^n}[g_{j'}^2]}}$ |
| **GENIE** | $\frac{1}{\mathbb{E}_{D \sim \mathcal{Z}^n}(g_j^2)}$ | $r_j + \frac{1}{n}$ | $1 - \frac{1}{n}\sum_{j \in J} W_j \cdot \frac{1}{\mathbb{E}_{j \in J}(r_j + \frac{1}{n})}$ | $W_j = \frac{1}{|J|}$ |

convergence speed.Incorporating both perspectives, Our method leads to a higher OSGR and thus improves generalization performance. Furthermore, we demonstrate that the alignment term in our preconditioning achieves a higher OSGR value than those of existing preconditioning methods. Detailed justifications are provided in the Appendix C.

### 3.3.2. GENERALIZATION ANALYSIS WITH PAC-BAYES BOUND

While the previous analysis is based on alignment and convergence dynamics using OSGR, we now adopt a complementary perspective grounded in the PAC-Bayes framework. We formulate the generalization analysis under a one-step update setting, where the KL divergence between successive parameter distributions reveals the connection between our preconditioning and a tighter generalization bound.

**Theorem 3.6** (PAC-Bayes Interpretation of Preconditioning). *$R(\theta)$ is the population risk and $L(\theta)$ is empirical risk. Assume that the loss function $L(\theta)$ is bounded in $[0, C]$. For any $\lambda > 0$, with probability at least $1 - \delta$ over the draw of $\mathcal{D}$, and for any data-dependent distribution $\tilde{p}$ over parameters $\theta$, the following PAC-Bayes bound holds:*

$$\mathbb{E}_{\theta \sim \tilde{p}}[R(\theta)] \leq \underbrace{\mathbb{E}_{\theta \sim \tilde{p}}[L(\theta)]}_{T_1} + \frac{\lambda C^2}{8n} + \underbrace{\frac{\mathrm{KL}(\tilde{p}\|\pi) + \log \frac{1}{\delta}}{\lambda}}_{T_2}.$$

*Assume that $\tilde{p} = \mathcal{N}(\theta_{t+1}, \Sigma_{\tilde{p}})$ and $\pi = \mathcal{N}(\theta_t, \Sigma_\pi)$, where $\Sigma_{\tilde{p}} = \mathrm{diag}(q_j \cdot \rho_j^2)$ and $\Sigma_\pi = \mathrm{diag}(\rho_j^2)$. Let $q_j = \frac{\mathbb{E}[g]^2}{\mathbb{E}[g_j^2]}$ be a variance adaptation factor from SVAG optimizer(Balles & Hennig, 2018) that minimizes the variance to reduce $\max T_1$ term. ($\theta_{t+1} = \theta_t - q \odot g$)*

*Then, minimizing the $T_2$ term via gradient descent yields an*

*update direction:*

$$\nabla_{\theta_t}\mathrm{KL}(\tilde{p}\|\pi) = \underbrace{\frac{1}{\mathbb{E}[g_j^2]} \cdot \frac{\mathbb{E}[g_j]^2}{\rho_j^2}}_{\text{GENIE}} \cdot g_{j,t},$$

*which matches the preconditioning rule of our optimizer.*

*Remark* 3.7 (Sharpness and Generalization via KL). This result shows that our method not only improves sharpness—as done in SAM—but also directly enhances generalization by minimizing both terms in the PAC-Bayes bound. Specifically, the variance adaptation factor $q_j$ reduces the variability of scaled gradients, thereby tightening the empirical loss term $T_1$ through more stable updates. Simultaneously, the $\frac{1}{\rho^2}$ term minimizes the KL divergence term $T_2$. This result shows our the generalization property of GENIE comes from correlation with Pac-Bayes theory.

### 3.3.3. CONVERGENCE ANALYSIS

This section analyzes the convergence properties of GENIE under non-convex settings. Specifically, we adopt three widely used assumptions in the optimization literature:

**Assumption 3.8.** (Bounded Gradient) There exists a constant $G > 0$ such that

$$\|\nabla\mathcal{L}(\theta_t)\| \leq G \quad \text{for all } t. \tag{9}$$

**Assumption 3.9.** (L-smooth) The loss function $\mathcal{L}$ is $L$-smooth, meaning there exists a constant $L > 0$ such that for all $\theta_1, \theta_2$:

$$\|\nabla\mathcal{L}(\theta_1) - \nabla\mathcal{L}(\theta_2)\| \leq L\|\theta_1 - \theta_2\|. \tag{10}$$

**Assumption 3.10.** (Lower bounded variance) The variance of the stochastic gradients have lower bound by a constant $1/S_u$:

$$\mathbb{E}[\|g_t - \nabla\mathcal{L}(\theta_t)\|^2] \geq 1/S_u, \quad \forall t. \tag{11}$$

Under these assumptions, we establish the following result regarding the convergence rate:

**Theorem 3.11.** *Under Assumption 3.8 Assumption 3.9, and Assumption 3.10 the average gradient norm over $T$ iterations can be expressed as:*

$$\mathbb{E}[\|\nabla\mathcal{L}(\theta)\|^2] \leq O\left(\frac{1}{P_l}\left(1 + \frac{G \cdot S_u^2}{2}\right)\frac{1}{\sqrt{\hat{T}}}\right). \quad (12)$$

*where $P_l$ is lower bound of preconditioning value.*

*Remark* 3.12 (Convergence Rate and Intuition). Theorem 3.11 shows that the average gradient norm converges at $O(T^{-1/2})$, the standard rate for stochastic gradient methods in non-convex optimization. This implies that GENIE retains the fundamental convergence properties of SGD.

*Remark* 3.13 (Influence of $G \cdot S_u$ and $S_u$). The term $G \cdot S_u^2$ represents a trade-off associated with the GSNR. A higher GSNR upper bound( $G \cdot S_u$) indicates a stronger gradient signal, which enhances generalization performance. However, it also acts as a multiplicative factor in the gradient norm, potentially slowing down convergence and thereby creating a trade-off. Furthermore, the variance term($S_u$) has a significant impact on the bound, further influencing the overall convergence behavior. To address this issue, we regulate the variance term using the $tanh$ function, which effectively balances the interplay between generalization and convergence dynamics.

## 4. Experiment

**Dataset.** We followed the standardized protocols of DomainBed (Gulrajani & Lopez-Paz, 2021), which include dataset splits, hyperparameter searches, and model selection using validation sets. Our approach was evaluated on five DG benchmark datasets: PACS (Li et al., 2017), VLCS (Fang et al., 2013), OfficeHome (Venkateswara et al., 2017), TerraIncognita (Beery et al., 2018), and DomainNet (Peng et al., 2019).

**Evaluation.** In accordance with DomainBed protocols, models were trained for 15,000 iterations on DomainNet and 5,000 iterations on the other datasets. For all DG and SDG experiments, we employed the Training-domain Validation Set approach, partitioning the source domain into training and validation subsets. The optimal model was selected based on validation performance. We followed previous DG methods by constructing 20 train-validation splits, with each split repeated 3 times.

**Implementation Details.** We used ResNet-50 (He et al., 2016b) pre-trained on ImageNet (He et al., 2016a) as backbone architectures. Detailed implementation details are presented in Appendix D. The detailed results and corresponding confidence intervals of all experiments are provided in Appendix E.

### 4.1. Comparison of Optimizers on DG

**Experiment Setup.** We examined the impact of various optimization methods on generalization performance under domain shifts using Baseline ERM (Vapnik, 1999). The evaluated methods included: Standard optimizers (SGD (Robbins & Monro, 1951)), Adaptive optimizers (Adam (Kingma, 2014), AdamW (Loshchilov & Hutter, 2019), AdaBelief (Zhuang et al., 2020), AdaHessian (Yao et al., 2021), YOGI (Zaheer et al., 2018)), Sharpness-aware optimizers (SAM (Foret et al., 2021), GAM (Zhang et al., 2023b), FAD (Zhang et al., 2023a)) and our proposed GENIE.

**Results.** As shown in Table 2, our optimizer achieved superior performance across most datasets, surpassing existing methods. GENIE outperformed Adam, the default optimizer in most DG algorithms(Zhang et al., 2023a), by 5.69%. Additionally, it achieved improvements of 6.36% over SGD and 4.37% over SAM. In particular, it achieved remarkable performance on VLCS, which is prone to early convergence and overfitting(Matsuura & Harada, 2020), and on TerraIncognita, a wildlife image dataset with significant challenges such as lighting variations, motion blur, occlusions, and severe class imbalance(Beery et al., 2018). These results suggest that GENIE effectively prevents overfitting and enhances the learning of causal relationships by balancing parameter contributions during training. Optimizers designed for generalization, such as SAM, GAM and FAD, outperform standard optimizers, underscoring the significant role of optimization in generalization. These results emphasize the need for developing optimizers specifically tailored for DG.

*Table 2.* Comparison of optimizers on DG datasets. Results denoted by * are reproduced from (Zhang et al., 2023a) using the same protocol as our paper. The best results for each dataset are highlighted in bold.

| OPT. | PACS | VLCS | OFFICE HOME | TERRA INC | DOMAIN NET | AVG. |
|---|---|---|---|---|---|---|
| ADAM* | 84.2 | 77.3 | 67.6 | 44.4 | 43.0 | 63.3 |
| ADAMW* | 83.6 | 77.4 | 68.8 | 45.2 | 43.4 | 63.7 |
| SGD* | 79.9 | 78.1 | 68.5 | 44.9 | 43.2 | 62.9 |
| YOGI* | 81.2 | 77.6 | 68.3 | 45.4 | 43.5 | 63.2 |
| ADABELIEF* | 84.6 | 78.4 | 68.0 | 45.2 | 43.5 | 63.9 |
| ADAHESSIAN* | 84.5 | 78.6 | 68.4 | 44.4 | **44.4** | 64.1 |
| SAM* | 85.3 | 78.2 | 68.0 | 45.7 | 43.4 | 64.1 |
| GAM* | 86.1 | 78.5 | 68.2 | 45.2 | 43.8 | 64.4 |
| FAD* | **88.2** | 78.9 | 69.2 | 45.7 | **44.4** | 65.3 |
| **GENIE** | 87.8 | **80.7** | **69.7** | **52.0** | 44.1 | **66.9** |

**Experiment Setup.** The computational overhead of an optimizer is a critical factor in its practical applicability. To evaluate this, we trained models on the PACS and VLCS

datasets for 5,000, 10,000, and 15,000 iterations, measuring average performance and training time per iteration.

**Results.** As reported in Table 3, GENIE consistently outperformed other optimizers, even at 5,000 iterations, while incurring lower computational overhead than SGD and Adam. Additionally, GENIE achieved an average of 1.3× faster training compared to SAM, as SAM's update rule requires two sequential (non-parallelizable) gradient computations per step, which doubles the training time. These results experimentally validate the theoretical convergence analysis in Section 3.3.3, confirming GENIE's ability in computational efficiency and convergence speed.

Table 3. Training time (sec) and average accuracy at different iteration levels.

| Opt. | Iter. | Training | | Avg. | |
|---|---|---|---|---|---|
| | | Time (/s) | PACS | VLCS | Office Home |
| SGD | 5000 | 5,273 | 69.8 | 76.7 | 51.3 |
| | 10000 | 10,546 | 73.9 | 77 | 62.5 |
| | 15000 | 15,783 | 75.8 | 77.7 | 63.9 |
| Adam | 5000 | 5,443 | 84.2 | 77 | 63.6 |
| | 10000 | 10,934 | 86.1 | 77 | 65.2 |
| | 15000 | 16,531 | 84.5 | 77 | 65.2 |
| SAM | 5000 | 5,775 | 82.4 | 79.4 | 69.4 |
| | 10000 | 11,500 | 83.5 | 80.3 | 69.6 |
| | 15000 | 17,191 | 84.1 | 80.4 | **70** |
| **GENIE** | 5000 | 4,292 | **88.4** | **81.3** | **70** |
| (ours) | 10000 | 8,582 | 87.1 | **81.3** | 69.2 |
| | 15000 | 12,876 | 86.9 | **81.3** | 69.1 |

### 4.2. Integration with Current DG Algorithms

**Experiment Setup.** GENIE is a versatile optimizer that integrates seamlessly with various DG algorithms without requiring changes to the training procedure or model architecture. To validate its compatibility, we combined GENIE with several well-performing DG algorithms—CORAL(Sun & Saenko, 2016) and RSC(Huang et al., 2020) using ResNet-50 as the backbone—and compared its performance against other optimization techniques.

**Results.** The performance evaluation results for DG are summarized in Table 5. GENIE consistently outperforms existing optimization methods, demonstrating its robustness and broad applicability. These results validate GENIE's scalability and compatibility with various DG algorithms. Unlike other DG methods, which often require multiple source domains or architecture modifications, GENIE seamlessly integrates with existing training pipelines, providing consistent performance gains without additional complexity. This establishes GENIE as an algorithm-agnostic and highly adaptable optimization framework for DG tasks.

### 4.3. Single Domain Generalization

**Experiment Setup.** We evaluated performance in Single Domain Generalization (SDG), which is more constrained but better reflects real-world applications. The flexibility to operate in SDG without structural modifications is an advantage of our method over certain existing methods that are limited to multi-source settings. In SDG, the model is trained and validated on a single domain and tested on the others, with results averaged across all source domains. We compared GENIE with Adam, SGD, and SAM, and applied it to existing DG methods.

**Results.** The SDG performance results are presented in Table 4. As in previous DG settings, our optimizer outperformed existing optimizers. When applied to DG methods, conventional optimizers reduced performance, whereas GENIE achieved the highest performance as a standalone model and also improved DG methods when used as an optimizer. These results show that our method enhances DG performance without requiring architectural modifications or multiple source domains, and performs well even as a standalone method.

Table 4. Experimental results of GENIE under the SDG setting.

| Algorithm | PACS | VLCS | Office Home | Terra Inc | Avg. |
|---|---|---|---|---|---|
| Adam | 64.3 | 56.2 | 50.7 | 33.5 | 51.2 |
| SGD | 49.5 | 60.4 | 45.9 | 22.8 | 44.7 |
| SAM | 57.7 | 66.7 | **59.2** | 26.8 | 52.6 |
| **GENIE (ours)** | **69.5** | **69.9** | 58.6 | **36.0** | **58.5** |
| RSC+Adam | 56.8 | 51.6 | 2.1 | 31.6 | 35.5 |
| RSC+SGD | 22.2 | 39.8 | 1.7 | 17.6 | 20.3 |
| **RSC+GENIE(ours)** | **68.2** | **68.7** | **54.4** | **33.2** | **56.1** |
| CORAL+Adam | 64.3 | 56.2 | 50.7 | 33.5 | 51.2 |
| CORAL+SGD | 49.5 | 60.4 | 45.9 | 22.8 | 44.7 |
| **CORAL+GENIE(ours)** | **70.9** | **69.2** | **56.4** | **36.7** | **58.3** |

### 4.4. Model Analysis

**Ablation.** We conducted an ablation study using the PACS dataset in a DG setting to evaluate the effects of Preconditioning, Noise Injection, and Random Mask (Table 6). The version without all three components corresponds to ERM trained with Adam, while the version incorporating all three represents our proposed GENIE optimizer. Experimental results show that GENIE achieved the highest performance, improving accuracy by 4.9% compared to ERM. Even when using only Preconditioning, performance improved by 3.8%, indicating that a simple preconditioning technique can enhance generalization. Additionally, in the Cartoon and Sketch domains, where objects are placed on a white background, models trained with Noise Injection and Random Mask performed better. Here, we conclude that preconditioning alone is enough for DG, but you can optionally utilize Noise Injection and Random Mask for

*Table 5.* Integration with DG methods. Results obtained from the original literature and DomainBed (Gulrajani & Lopez-Paz, 2021) are denoted with †, while results taken from (Zhang et al., 2023a) are denoted with *.

| ALGORITHM | PACS | VLCS | OFFICEHOME | TERRAINC | AVG. |
|---|---|---|---|---|---|
| ERM†(VAPNIK, 1999) | 85.5 | 77.5 | 66.5 | 46.1 | 68.9 |
| IRM†(ARJOVSKY ET AL., 2019) | 83.5 | 78.6 | 64.3 | 47.6 | 68.5 |
| GROUPDRO†(SAGAWA ET AL., 2020) | 84.4 | 76.7 | 66.0 | 43.2 | 67.6 |
| I-MIXUP†(XU ET AL., 2020) | 84.6 | 77.4 | 68.1 | 47.9 | 69.5 |
| MLDG†(LI ET AL., 2018A) | 84.9 | 77.2 | 66.8 | 47.8 | 69.2 |
| MMD†(LI ET AL., 2018B) | 84.7 | 77.5 | 66.4 | 42.2 | 67.7 |
| DANN†(GANIN ET AL., 2016) | 83.7 | 78.6 | 65.9 | 46.7 | 68.7 |
| CDANN†(LI ET AL., 2018C) | 82.6 | 77.5 | 65.7 | 45.8 | 67.9 |
| MTL†(BLANCHARD ET AL., 2021) | 84.6 | 77.2 | 66.4 | 45.6 | 68.5 |
| SAGNET†(NAM ET AL., 2021) | 86.3 | 77.8 | 68.1 | 48.6 | 70.2 |
| ARM†(ZHANG ET AL., 2021) | 85.1 | 77.6 | 64.8 | 45.5 | 68.3 |
| VREX†(KRUEGER ET AL., 2021) | 84.9 | 78.3 | 66.4 | 46.4 | 69 |
| MIXSTYLE*(ZHOU ET AL., 2021) | 85.2 | 77.9 | 60.4 | 44 | 66.9 |
| MIRO*(CHA ET AL., 2022) | 85.4 | 78.9 | 69.5 | 45.4 | 69.8 |
| **GENIE (OURS)** | **87.8** | **80.7** | **69.7** | **52.0** | **72.6** |
| RSC(HUANG ET AL., 2020)+ADAM* | 84.5 | 77.9 | 65.7 | 44.5 | 68.2 |
| RSC+ADAMW* | 83.4 | 77.5 | 66.3 | 45.1 | 68.1 |
| RSC+SGD* | 82.6 | 78.1 | 67 | 43.9 | 67.9 |
| **RSC+GENIE(OURS)** | **87.3** | **80.6** | **68.1** | **49.5** | **71.4** |
| CORAL(SUN & SAENKO, 2016) + ADAM* | 86 | 78.9 | 68.7 | 43.7 | 69.3 |
| CORAL+ADAMW* | 86.4 | 79.5 | 69.8 | 45.0 | 70.2 |
| CORAL+SGD* | 85.6 | 78.2 | 69.5 | 45.8 | 69.8 |
| **CORAL+GENIE(OURS)** | **87.9** | **80.7** | **70.6** | **48.4** | **71.9** |

additional robustness.

*Table 6.* Ablation study on the PACS dataset. Results are reported for evaluations on four domains: Art, Cartoon, Photo, and Sketch.

| PRE CONDITION | NOISE | MASK | PACS A | C | P | S | AVG. |
|---|---|---|---|---|---|---|---|
| X | X | X | 88.0 | 79.7 | 96.7 | 72.7 | 84.2 |
| O | X | X | **89.5** | 82.3 | 98.4 | 79.4 | 87.4 |
| O | O | X | 85.4 | 77.4 | 98.6 | 78.7 | 85.0 |
| O | X | O | 84.6 | 79.9 | 98.3 | 77.4 | 85.1 |
| O | O | O | 89.3 | **84.1** | **98.7** | **81.6** | **88.4** |

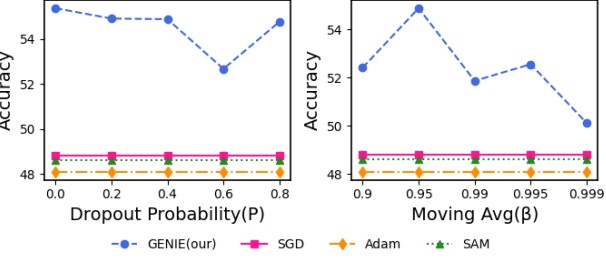

*Figure 2.* Performance sensitivity of GENIE to dropout probability $P$ and coefficient $B$.

**Sensitivity Analysis.** GENIE employs two key hyperparameters: the dropout probability $P$ and the coefficient $B$, which is used to compute the moving average and variance of gradients. To analyze the sensitivity of these hyperparameters, we conducted a grid search while keeping all other training settings fixed. As shown in Figure 2, GENIE consistently outperformed SGD, Adam, and SAM across a wide range of $P$ and $B$ values, demonstrating strong robustness to hyperparameter variation. Notably, while this experiment involved hyperparameter tuning via grid search, all other experiments followed the DomainBed protocol, using validation performance for hyperparameter selection.

**OSGR of Network Parameters Over Time.** To assess whether our approach enhances the overall OSGR of network parameters during training, we tracked the average OSGR of all parameters throughout the training process. As shown in Figure 4, the OSGR measurements on the VLCS dataset show that GENIE achieves an OSGR closer to 1 than prior optimizers. This means superior generalization performance. These findings align with the theoretical Generalization analysis in Section 3.3.1, confirming that GENIE ensures more stable and balanced parameter updates during training, which ultimately leads to improved generalization.

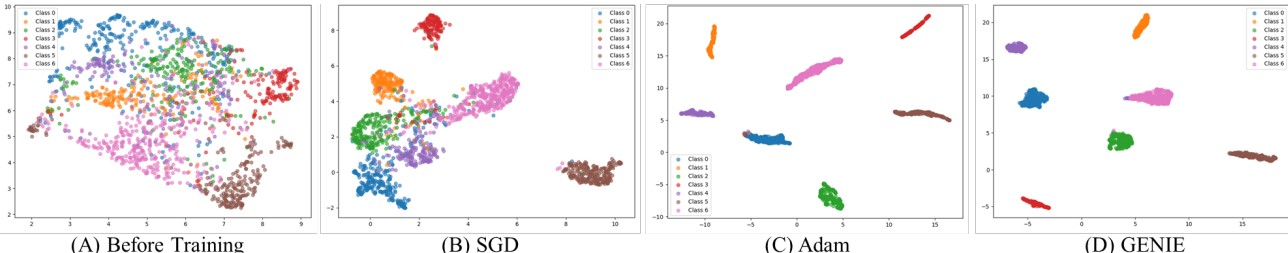

*Figure 3.* UMAP visualization of learned features on the PACS dataset with Sketch as the unseen target domain. (A) Before training. (B)–(D) After training with SGD, Adam, and GENIE.

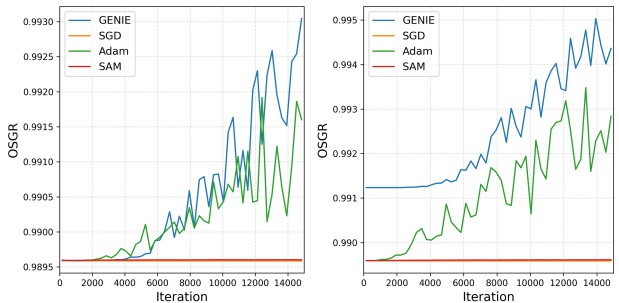

*Figure 4.* OSGR measurements over training iterations for different optimizers on the VLCS dataset. The OSGR values were calculated for all iterations and averaged every 200 iterations for clarity and ease of comparison.

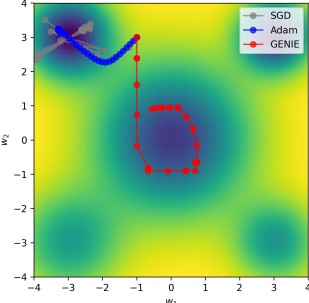

*Figure 5.* Optimization trajectories on a simulated loss landscape.

Interestingly, while SAM is designed for better generalization performance, it exhibits inferior OSGR values. This suggests that the sharpness-aware regime alone is insufficient for generalization, and that the OSGR regime should also be considered when addressing generalization in DG tasks. This observation is consistent with our PAC-Bayesian analysis in Section 3.3.2, which reveals that inducing balanced OSGR values leads to tighter generalization bounds, reinforcing the role of OSGR as a necessary complement to sharpness-aware optimization.

**Loss Landscape.** We analyzed the convergence paths of SGD, Adam, and GENIE in the loss landscape using the FashionMNIST dataset(Xiao et al., 2017). As shown in Figure 5, each corner represents the local minima of a specific source domain. All optimizers started at (-1,3) and were updated for 30 steps under the same conditions. SGD and Adam follow steep direction and converge quickly. However, fast convergence often causes overfitting to specific source domains in OOD scenarios. Generalizable features are learned later in training(Pérez et al., 2019; Shah et al., 2020b; Nakkiran et al., 2019), so rapid convergence can prevent the model from acquiring them sufficiently. In contrast, as demonstrated in the theoretical analysis in Section 3.3.2, GENIE leads optimization toward flatter minima by effec-

tively reducing sharpness, thereby improving generalization(Foret et al., 2021).

**Feature Visualization.** To examine how the GENIE optimizer operates at the feature level, we performed UMAP visualizations(McInnes et al., 2018) on the PACS dataset, with the Sketch domain held out as the unseen target. Each color represents a different class. The results Figure 3 show that GENIE leads to clear class separation across domains, suggesting effective domain-invariant feature learning.

## 5. Conclusion

We introduce GENIE, an optimizer that leverages OSGR to guide gradients in effective directions, preventing overly predictive parameters from dominating while ensuring all parameters contribute equitably to learning. GENIE achieves a higher OSGR with improved generalization and ensures fast convergence rate comparable to SGD. Empirically, it outperforms state-of-the-art optimizers across five DG benchmarks, demonstrating robust performance under significant domain shifts and limited data. Seamlessly integrating with existing DG and SDG methods, GENIE consistently achieves performance improvements. This work highlights the potential of OSGR as a guiding principle, paving the way for its use in few-shot learning, meta-learning, and other tasks requiring solutions to source-domain overfitting.

## Acknowledgements

This work was supported by Korea Internet & Security Agency(KISA) grant funded by the Korea government(PIPC) (No.RS-2023-00231200, Development of personal video information privacy protection technology capable of AI learning in an autonomous driving environment)

## Impact Statement

Our work proposes GENIE, an optimization method that enhances domain generalization by ensuring stable and balanced updates. GENIE mitigates overfitting, promotes flatter minima, and improves OOD performance, contributing to more robust and generalizable models.

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

# A. Notation

*Table 7.* Final Revised Notation Table

| Symbol | Description |
| --- | --- |
| $f$ | neural network |
| $L(\theta)$ | loss function |
| $\mathcal{Z}$ | data distribution defined over $\mathcal{X} \times \mathcal{Y}$ |
| $n$ | number of data samples |
| $D, D'$ | training/test dataset drawn from $\mathcal{Z}$ |
| $\theta, \theta_j$ | model parameters, parameter $j$ |
| $\theta_{t,j}$ | parameter of index $j$ at optimization step $t$ |
| $g_{D,j}(\theta)$ | gradient of parameter $j$ averaged over training set $D$ |
| $g_t$ | gradient at step $t$ |
| $g_j^2$ | squared gradient for parameter $j$ |
| $\rho_j^2$ | variance of parameter $j$'s gradient |
| $\sigma_j^2$ | variance of gradient averaged over training set |
| $r_j$ | gradient signal-to-noise ratio (GSNR), $r_j = \frac{g_j^2}{\rho_j^2}$ |
| $p_j$ | proposed preconditioning factor for parameter $j$ |
| $R(Z, n)$ | one-step generalization ratio (OSGR) |
| $\xi_t \sim \mathcal{N}(0, \sigma^2)$ | Gaussian noise for noise injection |
| $J$ | set of parameter index |
| $G$ | bound of gradient $l_2$ norm |
| $1/S_u$ | lower bound of gradient variance |
| $L$ | Lipschitz constant |
| $P_l$ | lower bound of preconditioning value |
| $W_j$ | weighting factor showing up in optimizers, $\frac{\mathbb{E}_{D \sim \mathcal{Z}^n}(g_{D,j}^2)}{\sum_{j'} \mathbb{E}_{D \sim \mathcal{Z}^n}(g_{D',j}^2)}$ in SGD |
| $\tilde{p}, \pi$ | probability measure of posterior and prior |
| $\boldsymbol{\Sigma}_{\tilde{p}}, \boldsymbol{\Sigma}_\pi$ | covariance matrix of Gaussian distribution |
| $\epsilon, \epsilon'$ | random error terms in gradients |
| $KL(\tilde{p}\|\pi)$ | KL divergence between distributions |

# B. Details of Table 1

We start out from a reinterpretation of the widely-used ADAM optimizer, which maintains moving averages of stochastic gradients and their element-wise square,

$$\tilde{m}_t = \beta_1 \tilde{m}_{t-1} + (1 - \beta_1)g_t, \quad \hat{m}_t = \frac{\tilde{m}_t}{1 - \beta_1^{t+1}}, \tag{13}$$

$$\tilde{v}_t = \beta_2 \tilde{v}_{t-1} + (1 - \beta_2)g_t^2, \quad \hat{v}_t = \frac{\tilde{v}_t}{1 - \beta_2^{t+1}}, \tag{14}$$

with $\beta_1, \beta_2 \in (0, 1)$ and updates with learning rate $\alpha$,

$$\theta_{t+1} = \theta_t - \alpha \frac{\hat{m}_t}{\sqrt{\hat{v}_t} + \varepsilon} \tag{15}$$

with a small constant $\varepsilon > 0$ preventing division by zero. Ignoring $\varepsilon$ and assuming $|m_{t,i}| > 0$ for the moment, we can rewrite the update direction as

$$\frac{m_t}{\sqrt{v_t}} = \text{sign}(m_t)\sqrt{\frac{v_t}{m_t^2}} = \underbrace{\frac{1}{\sqrt{1 + \frac{v_t - m_t^2}{m_t^2}}}}_{T_1} \circ \underbrace{\text{sign}(m_t)}_{T_2}. \tag{16}$$

Here, we divide the preconditioning into two terms. The convergnece term $T_2$ which modulates update size is :

$$\text{sign}(m_t) = \frac{\mathbb{E}_{D \sim \mathcal{Z}^n}(g_j)}{|\mathbb{E}_{D \sim \mathcal{Z}^n}(g_j)|} \tag{17}$$

The alignment term $T_1$ which includes GSNR is :

$$\frac{1}{\sqrt{1 + \frac{v_t - m_t^2}{m_t^2}}} = \sqrt{\frac{1}{\frac{1}{n \cdot r_j} + 1}} \tag{18}$$

The Equation (18) can be justified by definition of GSNR using Equation (19) and Equation (20). The variance of gradient average is:

$$v_t - m_t^2 = \frac{\rho_j^2}{n} \tag{19}$$

and

$$m_t^2 = \widetilde{g_j}^2 \tag{20}$$

## C. Proof of Theorems

### C.1. Convergence Analysis

We provide the detailed derivation and proof for Theorem 3.11. From Assumption 3.9, we start with:

$$L(\theta_{t+1}) \leq L(\theta_t) + \underbrace{\langle \nabla L(\theta_t), \theta_{t+1} - \theta_t \rangle}_{T_1} + \underbrace{\frac{L}{2}\|\theta_{t+1} - \theta_t\|^2}_{T2}, \tag{21}$$

where the first term, $T_1$, is given by:

$$T_1 = \langle \nabla L(\theta_t), \theta_{t+1} - \theta_t \rangle. \tag{22}$$

Using our preconditioning, which we defined as:

$$p = \frac{1}{\widetilde{g}^2 + \frac{\rho^2}{n}} \cdot \frac{\widetilde{g}^2}{\rho^2} = \frac{n}{n + \frac{\rho^2}{\widetilde{g}^2}} \cdot \frac{1}{\rho^2}, \tag{23}$$

which satysifies, given Assumption 3.10:

$$\frac{n}{n + \frac{\rho^2}{\widetilde{g}^2}} \leq 1, \frac{1}{\rho^2} \leq S_u \tag{24}$$

For $T_2$, we have:

$$T_2 = \frac{L}{2}\lambda^2\|p \odot g_t\|^2 \leq \frac{L}{2}\lambda^2\|S_u \cdot g_t\|^2, \tag{25}$$

and its expectation satisfies, using Assumption 3.8:

$$\mathbb{E}[T_2] \leq \frac{L}{2}\lambda^2 S_u^2 G^2. \tag{26}$$

For $T_1$, we decompose:
$$T_1 = \langle \nabla L(\theta_t), \theta_{t+1} - \theta_t \rangle = -\lambda_t \langle \nabla L(\theta_t), p \odot g_t \rangle, \tag{27}$$

$$T_1 \leq \underbrace{-\lambda_t P_l \langle \nabla L(\theta_t) \cdot g_t \rangle}_{T3} + \underbrace{\lambda_t \sum_j |[\nabla L(\theta_t)]_j| \cdot \frac{|g_{t,j}|}{\rho_{t,j}^2} \cdot 1(\text{sign}[[\nabla L(\theta_t)]_j] \neq \text{sign}[g_{t,j}])}_{T_4}. \tag{28}$$

Now, considering $T_3$, we evaluate its expectation:
$$\mathbb{E}[T_3] = -\lambda_t P_l \mathbb{E}[\langle \nabla L(\theta_t), g_t \rangle], \tag{29}$$

$P_l$ is lower bound of our preconditioning value. Considering $T_4$:

$$\mathbb{E}[T_4] = \lambda_t \sum_j \mathbb{E}\left[ |[\nabla L(\theta_t)]_j| \cdot \frac{|g_{t,j}|}{\rho_{t,j}^2} \cdot 1(\text{sign}[[\nabla L(\theta_t)]]_j \neq \text{sign}[g_{t,j}]) \right]. \tag{30}$$

Thus, we obtain:

$$\mathbb{E}[T_4] = \lambda_t \sum_j \mathbb{E}\left[ |[\nabla L(\theta_t)]_j| \cdot \frac{|g_{t,j}|}{\rho_{t,j}^2} \mid P(\text{sign}[[\nabla L(\theta_t)]_j] \neq \text{sign}[g_{t,j}]) \right]. \tag{31}$$

Next, we analyze the probability term:
$$P(\text{sign}[\nabla L(\theta_t)]_j \neq \text{sign}[g_{t,j}]) \tag{32}$$

and bound it as follows:
$$P(\text{sign}[\nabla L(\theta_t)]_j \neq \text{sign}[g_{t,j}]) \leq P(|[\nabla L(\theta_t)]_j - g_{t,j}| \geq |[\nabla L(\theta_t)]_j|). \tag{33}$$

Using Chebyshev's inequality:

$$P\big( |[\nabla L(\theta_t)]_j - g_{t,j}| \geq |[\nabla L(\theta_t)]_j| \big) \leq \frac{\text{Var}([\nabla L(\theta_t)]_j - g_{t,j})}{|[\nabla L(\theta_t)]_j|^2} = \frac{\sigma^2}{|[\nabla L(\theta_t)]_j|^2} = \frac{\rho_t^2/n}{|[\nabla L(\theta_t)]_j|^2}. \tag{34}$$

$n$ is the number of gradient samples.

Replacing the $n$ to step T, we bound the expectation:

$$\mathbb{E}[T_4] \leq \lambda_t \sum_j \mathbb{E}\left[ |[\nabla L(\theta_t)]_j| \cdot \frac{|g_{t,j}|}{\rho_{t,j}^2} \cdot \frac{\rho_{t,j}^2/T}{|[\nabla L(\theta_t)]_j|^2} \right] \leq \lambda_t \frac{|J|}{T} \tag{35}$$

$|J|$ is the number of parameters indicated by the size of parameter index set. Now, summing the inequalities until step $T$:

$$\mathbb{E}[L(\theta_{t+1})] \leq \mathbb{E}[L(\theta_t)] - \lambda_t P_l \|\nabla L(\theta_t)\|^2 + \lambda_t \cdot \frac{|J|}{T} + \frac{L}{2}\lambda_t^2 S_u^2 G^2. \tag{36}$$

Rearranging:

$$\mathbb{E}[L(\theta_{t+1})] \leq L(\theta_0) - \lambda_t P_l \sum_{t=1}^{T} \|\nabla L(\theta_t)\|^2 + T \cdot \lambda_t \left( \frac{|J|}{T} + \frac{L}{2}\lambda_t S_u^2 G^2 \right). \tag{37}$$

This results in:

$$\frac{1}{T}\sum_{t=1}^{T} \|\nabla L(\theta_t)\|^2 \leq \frac{L(\theta_0) - \mathbb{E}[L(\theta_{t+1})]}{\lambda_t P_l \cdot T} + \frac{1}{P_l}\left( \frac{|J|}{T} + \frac{L}{2}\lambda_t S_u^2 G^2 \right). \tag{38}$$

$$\frac{1}{T}\sum_{t=1}^{T} \|\nabla L(\theta_t)\|^2 \leq \frac{L(\theta_0) - \mathbb{E}[L(\theta_*)]}{\lambda_t P_l \cdot T} + \frac{1}{P_l}\left( \frac{|J|}{T} + \frac{L}{2}\lambda_t S_u^2 G^2 \right). \tag{39}$$

Taking $T \to \infty$, the convergence rate is:

$$\mathbb{E}[\|\nabla L(\theta_t)\|^2] \le \frac{L(\theta_0) - L(\theta_*)}{\lambda_t P_l \cdot T} + \frac{\frac{|J|}{T} + \frac{L}{2}\lambda_t S_u^2 G^2}{P_l}. \tag{40}$$

From the final steps of our derivation, we analyze the convergence rate of the algorithm.

Taking $\lambda_T$ as:

$$\lambda_T = \sqrt{\frac{L(\theta_0) - L(\theta_*)}{T \cdot \ell}}, \tag{41}$$

we have:

$$\mathbb{E}[\|\nabla L(\theta)\|] \le \frac{1}{P_l} \cdot \sqrt{\ell \frac{L(\theta_0) - L(\theta_*)}{T}} + \frac{G \cdot S_u^2}{2 \cdot P_l}\sqrt{\ell \frac{L(\theta_0) - L(\theta_*)}{T}} + \frac{|J|}{P_l \cdot T}. \tag{42}$$

Denoting $\frac{1}{\sqrt{\hat{T}}}$ as:

$$\frac{1}{\sqrt{\hat{T}}} = \sqrt{\frac{L(\theta_0) - L(\theta_*) \cdot \ell}{T}}, \tag{43}$$

Finally, we rewrite the bound, concluding that:

$$\mathbb{E}[\|\nabla L(\theta)\|^2] \le O\left(\frac{1}{P_l}\left(1 + \frac{G \cdot S_u^2}{2}\right)\frac{1}{\sqrt{\hat{T}}}\right) \quad \blacksquare \tag{44}$$

### C.2. Proof of Corollary 3.2

We utilized (Liu et al., 2020) for proving Corollary 3.2. In one gradient descent step, the model parameter is updated by $\Delta\theta = \theta_{t+1} - \theta_t = -\lambda p \odot g_D(\theta)$, where $\lambda$ is the learning rate and $p$ is preconditioning. If $\lambda$ is small enough, the one-step training and test loss decrease can be approximated by:

$$\Delta L[D] \approx -\Delta\theta \cdot \frac{\partial L[D]}{\partial \theta} + O(\lambda^2) = \lambda p \odot g_D(\theta) \cdot g_D(\theta) + O(\lambda^2), \tag{45}$$

$$\Delta L[D'] \approx -\Delta\theta \cdot \frac{\partial L[D']}{\partial \theta} + O(\lambda^2) = \lambda p \odot g_D(\theta) \cdot g_{D'}(\theta) + O(\lambda^2). \tag{46}$$

Usually, there are some differences between the directions of $g_D(\theta)$ and $g_{D'}(\theta)$, so statistically $\Delta L[D]$ tends to be larger than $\Delta L[D']$, and the generalization gap would increase during training. When $\lambda \to 0$, in one single training step, the empirical generalization gap increases by $\Delta L[D] - \Delta L[D']$. For simplicity, we denote this quantity as:

$$\nabla := \Delta L[D] - \Delta L[D'] \approx \lambda g_D(\theta) \cdot g_D(\theta) - \lambda g_D(\theta) \cdot g_{D'}(\theta), \tag{47}$$

which can be further simplified as:

$$\nabla = \lambda(p \odot \tilde{g}(\theta) + p \odot \epsilon)(\tilde{g}(\theta) + \epsilon') - \lambda(p \odot \tilde{g}(\theta) + p \odot \epsilon)(\tilde{g}(\theta) + \epsilon'), \tag{48}$$

$$\nabla = \lambda(\tilde{g}(\theta) + \epsilon)(\epsilon - \epsilon'). \tag{49}$$

Here, we replaced the random variables by $g_D(\theta) = \tilde{g}(\theta) + \epsilon$ and $g_{D'}(\theta) = \tilde{g}(\theta) + \epsilon'$, where $\epsilon$ and $\epsilon'$ are random variables with zero mean and variance $\sigma^2(\theta)$. Since $\mathbb{E}[\epsilon'] = \mathbb{E}[\epsilon] = 0$, $\epsilon$ and $\epsilon'$ are independent. The expectation of $\nabla$ is:

$$\mathbb{E}_{D,D'\sim\mathcal{Z}^n}(\nabla) = \mathbb{E}(\lambda p \odot \epsilon \cdot \epsilon') + O(\lambda^2) = \lambda \sum_j p_j \cdot \sigma^2(\theta_j) + O(\lambda^2), \tag{50}$$

where $\sigma^2(\theta_j)$ is the variance of the average gradient of the parameter $\theta_j$. For simplicity, when it involves a single model parameter $\theta_j$, we will use only a subscript $j$ instead of the full notation. For example, we use $\sigma_j^2$, $r_j$, and $g_{D,j}$ to denote $\sigma^2(\theta_j)$, $r(\theta_j)$, and $g_D(\theta_j)$, respectively.

**Expectation Analysis**

Consider the expectation of $\Delta L[D]$ and $\Delta L[D']$ when $\lambda \to 0$:

$$\mathbb{E}_{D \sim \mathcal{Z}^n}(\Delta L[D]) \approx \lambda \mathbb{E}_{D \sim \mathcal{Z}^n}(p \odot g_D(\theta) \cdot g_D(\theta)) = \lambda \sum_j p_j \mathbb{E}_{D \sim \mathcal{Z}^n}(g_{D,j}^2). \tag{51}$$

$$\mathbb{E}_{D,D' \sim \mathcal{Z}^n}(\Delta L[D']) = \mathbb{E}_{D,D' \sim \mathcal{Z}^n}(\Delta L[D] - \nabla) \approx \lambda \sum_j p_j (\mathbb{E}_{D \sim \mathcal{Z}^n}(g_{D,j}^2) - \sigma_j^2), \tag{52}$$

which simplifies further as:

$$\mathbb{E}_{D,D' \sim \mathcal{Z}^n}(\Delta L[D']) \approx \lambda \sum_j p_j (\mathbb{E}_{D \sim \mathcal{Z}^n}(g_{D,j}^2) - \rho_j^2/n). \tag{53}$$

**Simplification of $R(\mathcal{Z}, n)$**

Substituting Equation (53) and Equation (51) into $R(\mathcal{Z}, n)$, we have:

$$R(\mathcal{Z}, n) = 1 - \frac{\sum_j p_j \rho_j^2}{n \sum_j p_j \mathbb{E}_{D \sim \mathcal{Z}^n}(g_{D,j}^2)}. \tag{54}$$

When $r_j = \frac{\mathbb{E}_{D \sim \mathcal{Z}^n}(g_{D,j})^2}{\rho^2}$ We can rewrite Equation (53) as:

$$R(\mathcal{Z}, n) = 1 - \frac{1}{n} \sum_j \frac{p_j \mathbb{E}_{D \sim \mathcal{Z}^n}(g_{D,j}^2)}{\sum_{j'} p_j \mathbb{E}_{D \sim \mathcal{Z}^n}(g_{D',j}^2)} \cdot \frac{1}{r_j + \frac{1}{n}}, \tag{55}$$

or equivalently:

$$R(\mathcal{Z}, n) = \sum_j \frac{p_j \mathbb{E}_{D \sim \mathcal{Z}^n}(g_{D,j}^2)}{\sum_{j'} p_j \mathbb{E}_{D \sim \mathcal{Z}^n}(g_{D',j}^2)} \cdot \frac{1}{\left(1 + \frac{1}{n \cdot r_j}\right)}. \quad \blacksquare \tag{56}$$

## C.3. Generalization Analysis

### C.3.1. PAC BAYES BOUND AND PRECONDITIONING

**Theorem C.1** (PAC-Bayes bound). *Let $\mathcal{D} \sim \mathcal{Z}^n$ be a dataset sampled i.i.d. from a data distribution $\mathcal{Z}$. $R(\theta)$ is the population risk and $L(\theta)$ is empirical risk. Assume that the loss function $L(\theta)$ is bounded in $[0, C]$ for some constant $C > 0$. For any $\lambda > 0$, with probability at least $1 - \delta$ over the draw of $\mathcal{D}$, and for any data-dependent distribution $\tilde{p}$ over parameters $\theta$, the following PAC-Bayes bound holds:*

$$\mathbb{E}_{\theta \sim \tilde{p}}[R(\theta)] \leq \underbrace{\mathbb{E}_{\theta \sim \tilde{p}}[L(\theta)]}_{T_1} + \frac{\lambda C^2}{8n} + \underbrace{\frac{\mathrm{KL}(\tilde{p} \| \pi) + \log \frac{1}{\delta}}{\lambda}}_{T_2}.$$

Here, $R(\theta)$ denotes the expected loss over the true data distribution. The SAM algorithm primarily focuses on minimizing the $T_1$ term in Theorem C.1, following the inequality:

$$L_{\mathcal{D}}(\theta) \leq \mathbb{E}_{\epsilon \sim \mathcal{N}(0, \rho)}[L_{\mathcal{D}}(\theta + \epsilon)] \leq \max_{\|\epsilon\|_2 \leq \rho}[L_{\mathcal{D}}(\theta + \epsilon)]. \tag{57}$$

In contrast, our method simultaneously minimizes both the $T_1$ and $T_2$ terms. First, we determine a preconditioning vector $q$ to reduce $\mathbb{E}_S \mathbb{E}_{\theta \sim \tilde{p}}[L(\theta)]$. To minimize variance-induced error, we adopt the variance adaptation factor introduced in the SVAG optimizer(Balles & Hennig, 2018), and solve:

$$\mathbb{E}[\|q \odot g - \mathbb{E}[g]\|_2^2] = \sum_j q_j^2 \mathbb{E}[g_j^2] - 2q_j \mathbb{E}[g]^2 + \mathbb{E}[g]^2. \tag{58}$$

Minimizing this expression yields the optimal preconditioning:

$$q_j = \frac{\mathbb{E}[g]^2}{\mathbb{E}[g_j^2]}.$$

Next, assume $\tilde{p} = \mathcal{N}(\theta_{t+1}, \boldsymbol{\Sigma}_{\tilde{p}})$ and $\pi = \mathcal{N}(\theta_t, \boldsymbol{\Sigma}_{\pi})$, with both covariances defined as:

$$\boldsymbol{\Sigma}_{\tilde{p}} = \mathrm{diag}(q_1 \cdot \rho_1^2, q_2 \cdot \rho_2^2, \ldots, q_{|J|} \cdot \rho_{|J|}^2), \quad \boldsymbol{\Sigma}_{\pi} = \mathrm{diag}(\rho_1^2, \rho_2^2, \ldots, \rho_{|J|}^2),$$

Here, the prior $\pi$ can be treated as a data-driven prior which is approximated with stochastic gradient descent using all data excluding the current mini-batch. Assuming the variances do not significantly change between steps. Then the KL divergence can be written as follows:

$$KL(\tilde{p}\|\pi) = \frac{1}{2}\left[\sum_{i \in J}\frac{q_i \cdot \rho_i^2}{\rho_i^2} + \sum_{i \in J}\frac{(\theta_{t+1,i} - \theta_{t,i})^2}{\rho_i^2} - |J| + \sum_{i \in J}\log\left(\frac{\rho_i^2}{q_i \cdot \rho_i^2}\right)\right] \tag{59}$$

The gradient of this KL term with respect to $\theta_t$ is:

$$[\nabla_{\theta_t}KL(\tilde{p}\|\pi)]_j = -\frac{(\theta_{t+1,j} - \theta_{t,j})}{\rho_j^2} = \frac{\mathbb{E}[g_j]^2}{\mathbb{E}[g_j^2]} \cdot \frac{g_{j,t}}{\rho_j^2} = \underbrace{\frac{1}{\mathbb{E}[g_j^2]} \cdot \frac{\mathbb{E}[g_{j,t}]^2}{\rho_j^2}}_{\text{GENIE}} \cdot g_{j,t}. \tag{60}$$

This shows that minimizing the KL divergence in the PAC-Bayes bound via gradient descent naturally leads to the same preconditioning structure. Therefore, our method considers both sharpness (through variance adaptation) and generalization (via KL divergence minimization), whereas SAM focuses solely on sharpness.

It is important to note that our gradient computation is taken with respect to the prior mean $\theta_t$, rather than the posterior mean $\theta_{t+1}$. Due to the asymmetric nature of the forward KL divergence $KL(\tilde{p}\|\pi)$, this choice induces a *mode-covering* behavior rather than mode-seeking. As optimization proceeds, the prior distribution $\pi$ is iteratively adapted to cover a broader region of the risk landscape, effectively reducing over-concentration around a single mode. This mechanism aligns with the argument in *Risk Extrapolation* (Krueger et al., 2021), where covering a wider set of hypotheses can improve out-of-distribution generalization. Consequently, although the PAC-Bayes bound is originally derived under an i.i.d. assumption, this mode-covering property enables the prior to capture more diverse risk regions, thereby enhancing robustness under distribution shift.

### C.3.2. OSGR BASED ANALYSIS

From Section 3.2.1, the OSGR of our method is given by:

$$R_{\text{ours}} = 1 - \frac{1}{n}\sum_j\frac{1}{\sum_{j'}\left(r_{j'} + \frac{1}{n}\right)} = 1 - \frac{1}{n\mathbb{E}_{j\sim J}\left(r_j + \frac{1}{n}\right)}. \tag{61}$$

Similarly, from Theorem 3.1, the OSGR of SGD is given by:

$$R_{\text{sgd}} = 1 - \frac{1}{n}\sum_j\frac{\mathbb{E}_{D\sim\mathcal{Z}^n}[g_j^2]}{\sum_{j'}\mathbb{E}_{D\sim\mathcal{Z}^n}[g_{j'}^2]} \cdot \frac{1}{r_j + \frac{1}{n}}. \tag{62}$$

Replacing the term $\sum_j\frac{\mathbb{E}_{D\sim\mathcal{Z}^n}[g_j^2]}{\sum_{j'}\mathbb{E}_{D\sim\mathcal{Z}^n}[g_{j'}^2]}$ to $\sum W_j = 1$ which represents a weighted average, we rewrite it as:

$$R_{\text{sgd}} = 1 - \frac{1}{n}\sum_{j \in J}W_j\left(\frac{1}{r_j + \frac{1}{n}}\right). \tag{63}$$

If we assume uniform weight $W_j = \frac{1}{|J|}$, by Jensen's inequality:

$$0 \le 1 - \frac{1}{n} \sum_{j \in J} W_j \left( \frac{1}{r_j + \frac{1}{n}} \right) \le 1 - \frac{1}{n \mathbb{E}_{j \in J} \left( r_j + \frac{1}{n} \right)} \le 1. \tag{64}$$

Thus, we conclude:

$$0 \le R_{\text{sgd}} \le R_{\text{ours}} \le 1. \quad \blacksquare \tag{65}$$

In the same way, with any preconditioning,

$$R(\mathcal{Z}, n) = 1 - \frac{1}{n} \sum_j \frac{p_j \mathbb{E}_{D \sim \mathcal{Z}^n}(g_{D,j}^2)}{\sum_{j'} p_j \mathbb{E}_{D \sim \mathcal{Z}^n}(g_{D',j}^2)} \cdot \frac{1}{r_j + \frac{1}{n}}, \tag{66}$$

Replacing the term $\sum_j \frac{p_j \mathbb{E}_{D \sim \mathcal{Z}^n}(g_{D,j}^2)}{\sum_{j'} p_j \mathbb{E}_{D \sim \mathcal{Z}^n}(g_{D',j}^2)}$ to $\sum W_j = 1$ with $W_j = \frac{1}{|J|}$, which represents a average, we rewrite it as:

$$R_{precondition} = 1 - \frac{1}{n} \sum_{j \in J} W_j \left( \frac{1}{r_j + \frac{1}{n}} \right). \tag{67}$$

Also by Jensen's inequality, we obtain:

$$0 \le R_{\text{precondition}} \le R_{\text{ours}} \le 1. \quad \blacksquare \tag{68}$$

Consequently, this result establishes that our method achieves the highest OSGR value among preconditioning methods such as Adam, RMSprop, and SVAG(Balles & Hennig, 2018).

However, the assumption of uniform weights $W_j = \frac{1}{|J|}$ can be overly restrictive. In practice, preconditioning methods aim to balance the contribution of parameter updates, which becomes especially important when there exists a strong imbalance in the GSNR distribution. We consider the case where a dominant coordinate exists, denoted as $j_{\max} = \arg\max_{j \in J} r_j$, and define the remaining coordinates as $J' = J \setminus \{j_{\max}\}$, with $j'_{\max} = \arg\max_{j \in J'} r_j$, such that $r_{j'_{\max}} \ll r_{j_{\max}}$.

To demonstrate that our method still yields a higher OSGR under such imbalance, we consider the difference:

$$n(R_{\text{ours}} - R_{\text{precondition}}) = \sum_{j \in J} W_j \left( \frac{1}{r_j + \frac{1}{n}} \right) - \frac{1}{\mathbb{E}_{j \in J} \left( r_j + \frac{1}{n} \right)}. \tag{69}$$

For sufficiently large $n$, the $\frac{1}{n}$ terms can be neglected:

$$n(R_{\text{ours}} - R_{\text{precondition}}) \approx \sum_{j \in J} W_j \left( \frac{1}{r_j} \right) - \frac{1}{\mathbb{E}_{j \in J}(r_j)}. \tag{70}$$

Separating the contribution of the dominant coordinate $j_{\max}$, we have:

$$n(R_{\text{ours}} - R_{\text{precondition}}) = \sum_{j \in J'} W_j \left( \frac{1}{r_j} \right) + W_{j_{\max}} \left( \frac{1}{r_{j_{\max}}} \right) - \frac{1}{\mathbb{E}_{j \in J}(r_j)}. \tag{71}$$

Since $\mathbb{E}_{j \in J}(r_j) \ge \frac{r_{j_{\max}}}{|J|}$, this difference is lower-bounded by:

$$n(R_{\text{ours}} - R_{\text{precondition}}) \ge \sum_{j \in J'} W_j \left( \frac{1}{r_j} \right) + W_{j_{\max}} \left( \frac{1}{r_{j_{\max}}} \right) - \frac{|J|}{r_{j_{\max}}}. \tag{72}$$

Further bounding the terms using $r_j \le r_{j'_{\max}}$ for all $j \in J'$, we obtain:

$$\sum_{j \in J'} W_j \left( \frac{1}{r_j} \right) + W_{j_{\max}} \left( \frac{1}{r_{j_{\max}}} \right) - \frac{|J|}{r_{j_{\max}}} \ge (1 - W_{j_{\max}}) \left( \frac{1}{r_{j'_{\max}}} \right) + W_{j_{\max}} \left( \frac{1}{r_{j_{\max}}} \right) - \frac{|J|}{r_{j_{\max}}}. \tag{73}$$

Therefore, if the following condition holds:

$$(1 - W_{j_{\max}})r_{j_{\max}} \geq |J|r_{j'_{\max}}, \tag{74}$$

then $R_{\text{ours}} \geq R_{\text{precondition}}$, and our method guarantees superior OSGR performance compared to other preconditioning strategies. This result highlights the robustness of our formulation, particularly under skewed GSNR distributions.

# D. Implementation Details

### D.1. Training details

As introduced in the experimental section, we follow the standard training, hyperparameter search methods, and evaluation protocol proposed by DomainBed (Gulrajani & Lopez-Paz, 2021) to ensure a fair comparison. For each dataset, the models were trained for 15,000 iterations on DomainNet and 5,000 iterations on the other datasets. The search space of hyperparameters is provided in Table 8. All experiments were conducted on an NVIDIA GeForce RTX 4090 under the environment of Python 3.8.10, PyTorch 1.13.1, Torchvision 0.14.1, and CUDA 11.7.

*Table 8.* The search space of hyperparameters.

| PARAMETER | DEFAULT VALUE | SEARCH DISTRIBUTION |
|---|---|---|
| BATCH SIZE | 32 | $2^{\text{UNIFORM}(3,5.5)}$ |
| LEARNING RATE | 0.015 | $10^{\text{UNIFORM}(-3,-1)}$ |
| RESNET DROPOUT | 0.0 | $[0.0, 0.1, 0.5]$ |
| WEIGHT DECAY | 0.0 | $10^{\text{UNIFORM}(-6,-2)}$ |

## D.2. Pseudo code

---

**Algorithm 2** Unified Algorithm for RMSProp, Adam, and GENIE

---

**Input:** Mini-batches $\{\mathcal{B}_t\}_{t=1}^T$, learning rate $\alpha$, total steps $T$

**Preconditioning Parameters:** $\beta, \beta_1, \beta_2 \in [0,1]$, noise scale $\sigma$, $m_0 \leftarrow 0$, $v_0 \leftarrow 0$

**Initialize:** Parameters $\theta_0$

**for** $t = 1$ **to** $T$ **do**

    ▷ **Compute Gradient:** Sample mini-batch $\mathcal{B}_t$ and compute

$$g_t = \nabla \mathcal{L}(\theta_t; \mathcal{B}_t).$$

    ▷ **Compute Preconditioned Gradient:**

    **RMSProp:**

$$v_t \leftarrow \beta v_{t-1} + (1 - \beta)g_t^2, \quad \tilde{g}_t \leftarrow \frac{g_t}{\sqrt{v_t} + \epsilon}.$$

    **Adam:**

$$m_t \leftarrow \beta_1 m_{t-1} + (1 - \beta_1)g_t, \quad v_t \leftarrow \beta_2 v_{t-1} + (1 - \beta_2)g_t^2,$$

$$\hat{m}_t \leftarrow \frac{m_t}{1 - \beta_1^t}, \quad \hat{v}_t \leftarrow \frac{v_t}{1 - \beta_2^t}, \quad \tilde{g}_t \leftarrow \frac{\hat{m}_t}{\sqrt{\hat{v}_t} + \epsilon}.$$

    **GENIE:**

$$m_t \leftarrow \beta m_{t-1} + (1 - \beta)g_t, \quad v_t \leftarrow \beta v_{t-1} + (1 - \beta)g_t^2,$$

$$\sigma_t^2 = v_t - m_t^2, \quad \mathrm{r}_t = \tanh(\frac{1}{\sigma_t^2})m_t^2$$

$$\hat{g}_t \leftarrow \frac{m_t}{1 - \beta^t} \cdot \frac{1}{v_t} \cdot \mathrm{r}_t,$$

$$\mathrm{Noise}_t \leftarrow \xi_t \big[1 - \tanh(\frac{1}{\sigma_t^2})\big], \quad \xi_t \sim \mathcal{N}(0, \sigma^2),$$

$$\tilde{g}_t \leftarrow \hat{g}_t + \mathrm{Noise}_t.$$

$$\tilde{g}_t \leftarrow \tilde{g}_t \odot M, \ M_j \sim \mathrm{Bernoulli}(p)$$

    ▷ **Update Parameters:**

$$\theta_{t+1} \leftarrow \theta_t - \alpha \tilde{g}_t.$$

**end for**

**Output:** Final parameters $\theta_{T+1}$ for RMSProp, Adam, and GENIE.

---

## D.3. Code

```
[caption={Python implementation of preconditioning updates.}, label={lst:preconditioning}]
def _initialize_preconditioning(self, current_state):
    self.prev_state = current_state
    self.gmean = {k: torch.zeros_like(param) for k, param in
        self.network.named_parameters()}
    self.ge2 = {k: torch.zeros_like(param) for k, param in
        self.network.named_parameters()}
    self.scale = 0.0

def _update_preconditioning(self, lr, moving_avg):
    grad_sgd = {}
    pgrad = {}
    pGsnr = {}

    # Update scale factors
```

```
14      self.scale = (moving_avg * self.scale + 1.0)
15      scale1 = (1 - moving_avg) * self.scale
16      scale2 = 2.0 - scale1
17      rho = (1.0 - moving_avg) * scale2 / ((1.0 + moving_avg) * scale1)
18
19      with torch.no_grad():
20          # Update gradients and variance
21          for k, param in self.network.named_parameters():
22              delta = param.grad.data.detach()
23              self.gmean[k] = self.gmean[k] * moving_avg + delta * (1.0 - moving_avg)
24              self.ge2[k] = self.ge2[k] * moving_avg + (delta ** 2) * (1.0 - moving_avg)
25
26              gm = self.gmean[k] / scale1
27              ge2 = self.ge2[k] / scale1
28              var = ge2 - gm.square()
29              var /= (1.0 - rho)
30              var = torch.where(var > 0.0, var, torch.zeros_like(var) + 1e-8)
31
32              invvar = torch.clamp(1 / var, min=0.0, max=10.0)
33              mvar = rho * var
34              mvar = torch.where(mvar > 0.0, mvar, torch.zeros_like(mvar) + 1e-8)
35
36              # Preconditioned gradient scaling
37              tanh_invvar = torch.tanh(invvar)
38              pGsnr[k] = (1.0 / (1.0 + mvar / (gm.square() + 1e-8))) * tanh_invvar
39
40              # Add noise for stochastic gradient adjustment
41              noise_scale = torch.sum(tanh_invvar * torch.abs(gm) *
42                                      (1.0 / (1.0 + mvar / (gm.square() + 1e-8)))) /
43                                          torch.sum(tanh_invvar)
43              noise = torch.normal(torch.zeros_like(delta), torch.ones_like(delta)) *
                    noise_scale
44              grad_sgd[k] = (1 - tanh_invvar) * noise
45
46          # Compute preconditioned gradients
47          for k, param in self.network.named_parameters():
48              pgrad[k] = self.gmean[k] / scale1 * pGsnr[k].view_as(param)
49
50          # Apply gradients with dropout-based masking
51          for k, param in self.network.named_parameters():
52              mask = (torch.rand_like(param) > self.hparams['p']).float() / (1 -
                    self.hparams['p'])
53              self.prev_state[k] -= (pgrad[k] + grad_sgd[k]) * mask * lr
```

# E. Experimental Details and Results

### E.1. Dataset

This section introduces five representative DG datasets utilized in this paper.

- **PACS** (Li et al., 2017): This dataset includes 4 different domain styles—Photo, Art Painting, Cartoon, and Sketch. Each domain contains 7 categories and consists of 9,991 images. It is well-suited for evaluating generalization performance across style variations.

- **OfficeHome** (Venkateswara et al., 2017): This dataset consists of 4 different domain styles—Art, Clipart, Product, and Real-world. Each domain includes 65 categories, with a total of 15,588 samples.

- **VLCS** (Fang et al., 2013): Derived from four distinct datasets—Caltech101, LabelMe, VOC2007, and SUN09—this dataset includes 5 shared classes across domains, containing a total of 10,729 images. It is ideal for evaluating distributional differences between datasets.

- **Terra Incognita** (Beery et al., 2018): Comprising photographs of wildlife, this dataset is collected from 4 different

locations—L100, L38, L43, and L46. It includes 10 categories and 24,788 samples and is commonly used to measure model generalization performance in real-world scenarios.

- DomainNet (Peng et al., 2019): This large-scale dataset includes 6 domains—Clipart, Infograph, Painting, Quickdraw, Real, and Sketch. It comprises 345 categories and a total of 586,575 samples.

### E.2. Detailed Results: Comparison with Existing Optimizers

We compare the performance of existing optimizers with our proposed GENIE optimizer on five datasets from the same DG benchmark. The dataset-specific experimental results are presented in Table 9, Table 10, Table 11, Table 12, and Table 13. Additionally, we analyze the performance and training time as the number of iterations increases, as shown in Table 14, Table 15, Table 16.

*Table 9.* Comparison Results on the PACS Dataset.

| OPTIMIZER | ART | CARTOON | PHOTO | SKETCH | AVG. |
|---|---|---|---|---|---|
| ADAM* | 88.0±1.2 | 79.7±0.5 | 96.7±0.4 | 72.7±0.9 | 84.3 |
| ADAMW* | 84.1±1.5 | 80.7±1.2 | 96.9±0.4 | 72.8±0.6 | 83.6 |
| SGD* | 85.1±0.4 | 76.0±0.3 | 98.3±0.4 | 60.3±6.1 | 79.9 |
| YOGI* | 84.4±1.7 | 79.7±0.6 | 95.8±0.3 | 65.1±1.5 | 81.2 |
| ADABELIEF* | 85.4±2.2 | 80.4±1.1 | 97.4±0.7 | 75.1±1.4 | 84.6 |
| ADAHESSIAN* | 88.4±0.6 | 80.0±0.9 | 97.7±0.4 | 71.7±4.1 | 84.5 |
| SAM* | 85.7±1.2 | 81.0±1.4 | 97.1±0.2 | 77.4±1.8 | 85.3 |
| GAM* | 85.9±0.9 | 81.3±1.6 | 98.2±0.4 | 79.0±2.1 | 86.1 |
| FAD* | 88.5±0.5 | **83.0±0.8** | 98.4±0.2 | **82.8±0.9** | **88.2** |
| GENIE (OURS) | **88.7±0.7** | 82.8±1.3 | **98.5±0.1** | 81.3±0.4 | 87.8 |

*Table 10.* Comparison Results on the VLCS Dataset.

| OPTIMIZER | CALTECH | LABELME | SUN | VOC | AVG. |
|---|---|---|---|---|---|
| ADAM* | 98.9±0.4 | 65.9±1.5 | 71.0±1.6 | 74.5±2.0 | 77.3 |
| ADAMW* | 98.3±0.1 | 65.1±1.7 | 70.9±1.3 | 75.2±1.5 | 77.4 |
| SGD* | 98.4±0.2 | 64.7±0.7 | 72.5±0.8 | 76.6±0.8 | 78.1 |
| YOGI* | 98.1±0.7 | 63.9±1.2 | 72.5±1.6 | 75.7±1.2 | 77.6 |
| ADABELIEF* | 98.0±0.1 | 63.9±0.4 | 73.4±1.0 | 78.2±1.8 | 78.4 |
| ADAHESSIAN* | 99.1±0.3 | 65.0±1.7 | 72.7±1.3 | 77.7±1.0 | 78.6 |
| SAM* | 98.5±1.0 | 66.2±1.6 | 72.0±1.0 | 76.1±1.0 | 78.2 |
| GAM* | 98.8±0.6 | 65.1±1.2 | 72.9±1.0 | 77.2±1.9 | 78.5 |
| FAD* | 99.1±0.5 | 66.8±0.9 | 73.6±1.0 | 76.1±1.3 | 78.9 |
| GENIE (OURS) | **99.3±0.3** | **67.2±1.5** | **76.6±0.3** | **79.7±0.8** | **80.7** |

*Table 11.* Comparison Results on the OfficeHome Dataset.

| OPTIMIZER | ART | CLIPART | PRODUCT | REAL-WORLD | AVG. |
|---|---|---|---|---|---|
| ADAM* | 63.9±0.8 | 48.1±0.6 | 77.0±0.9 | 81.8±1.6 | 67.6 |
| ADAMW* | 66.1±0.7 | 48.7±0.6 | 76.6±0.8 | 83.6±0.4 | 68.8 |
| SGD* | 65.3±0.8 | 48.8±1.4 | 76.7±0.3 | 83.0±0.7 | 68.5 |
| YOGI* | 63.5±1.0 | 49.2±1.2 | 76.2±0.5 | 84.5±0.6 | 68.3 |
| ADABELIEF* | 65.6±2.0 | 48.1±0.9 | 74.8±0.8 | 83.6±0.9 | 68 |
| ADAHESSIAN* | 63.0±2.9 | 50.0±1.4 | 77.7±0.8 | 83.0±0.5 | 68.4 |
| SAM* | 63.5±1.2 | 48.6±0.9 | 77.0±0.8 | 82.9±1.3 | 68 |
| GAM* | 63.0±1.2 | 49.8±0.5 | 77.6±0.6 | 82.4±1.0 | 68.2 |
| FAD* | 63.5±1.0 | 50.3±0.8 | **78.0±0.4** | **85.0±0.6** | 69.2 |
| GENIE (OURS) | **66.2±0.5** | **55.0±0.4** | 77.5±0.4 | 80.0±0.5 | **69.7** |

*Table 12.* Comparison Results on the TerraIncognita Dataset.

| OPTIMIZER | L100 | L38 | L43 | L46 | AVG. |
|---|---|---|---|---|---|
| ADAM* | 42.2±3.4 | 40.7±1.2 | 59.9±0.2 | 35.0±2.8 | 44.4 |
| ADAMW* | 44.2±6.8 | 39.8±1.9 | 60.3±2.0 | 36.6±1.8 | 45.2 |
| SGD* | 41.8±5.8 | 39.8±3.9 | 60.5±2.2 | 37.5±1.1 | 44.9 |
| YOGI* | 43.9±2.2 | 42.5±2.6 | 60.5±1.1 | 34.8±1.6 | 45.4 |
| ADABELIEF* | 42.6±6.7 | 43.0±2.0 | 60.2±1.3 | 35.1±0.3 | 45.2 |
| ADAHESSIAN* | 42.5±4.8 | 39.5±1.0 | 58.4±2.6 | 37.3±0.8 | 44.4 |
| SAM* | 42.9±3.5 | 43.0±2.2 | 60.5±1.6 | 36.4±1.2 | 45.7 |
| GAM* | 42.2±2.6 | 42.9±1.7 | 60.2±1.8 | 35.5±0.7 | 45.2 |
| FAD* | 44.3±2.2 | 43.5±1.7 | **60.9±2.0** | 34.1±0.5 | 45.7 |
| GENIE (OURS) | **55.2 ± 4.8** | **47.5± 2.1** | 59.2± 0.4 | **45.9± 1.0** | **52.0** |

*Table 13.* Comparison Results on the DomainNet Dataset.

| OPTIMIZER | CLIP | INFO | PAINT | QUICK | REAL | SKETCH | AVG. |
|---|---|---|---|---|---|---|---|
| ADAM* | 63.0±0.3 | 20.2±0.4 | 49.1±0.1 | 13.0±0.3 | 62.0±0.4 | 50.7±0.1 | 43.0 |
| ADAMW* | 63.0±0.6 | 20.6±0.2 | 49.6±0.0 | 13.0±0.2 | 63.6±0.2 | 50.4±0.1 | 43.4 |
| SGD* | 61.3±0.2 | 20.4±0.2 | 49.4±0.2 | 12.6±0.1 | **65.7±0.0** | 49.6±0.2 | 43.2 |
| YOGI* | 63.3±0.1 | 20.6±0.1 | 50.1±0.3 | 13.2±0.3 | 62.8±0.1 | 51.0±0.2 | 43.5 |
| ADABELIEF* | 63.5±0.2 | 20.5±0.1 | 50.0±0.3 | 13.2±0.3 | 63.1±0.1 | 50.7±0.1 | 43.5 |
| ADAHESSIAN* | 63.3±0.2 | 21.4±0.1 | 50.8±0.3 | 13.6±0.1 | **65.7±0.1** | 51.4±0.2 | 44.4 |
| SAM* | 63.3±0.1 | 20.3±0.3 | 50.0±0.3 | 13.6±0.2 | 63.6±0.3 | 49.6±0.4 | 43.4 |
| GAM* | 63.0±0.5 | 20.2±0.2 | 50.3±0.1 | 13.2±0.3 | 64.5±0.2 | 51.6±0.5 | 43.8 |
| FAD* | **64.1±0.3** | **21.9±0.2** | **50.6±0.3** | **14.2±0.4** | 63.6±0.1 | 52.2±0.2 | **44.4** |
| GENIE (OURS) | 62.5±0.5 | 21.3±0.4 | 50.0±0.4 | 14.0±0.4 | 64.0±0.7 | **52.6±0.8** | 44.1 |

*Table 14.* Comparison of Optimizers on PACS Dataset Across Iterations.

| OPTIMIZER | ITERATION | TRAINING TIME (/S) | | | | ACCURACY | | | | AVG. | |
|---|---|---|---|---|---|---|---|---|---|---|---|
| | | [0] | [1] | [2] | [3] | [0] | [1] | [2] | [3] | TIME | ACC |
| SGD | 5000 | 1785 | 1819 | 1823 | 1848 | 73.4 | 61.2 | 96 | 48.4 | 1819 | 69.8 |
| | 10000 | 3570 | 3643 | 3646 | 3749 | 76.8 | 65.1 | 97.4 | 56.1 | 3652 | 73.9 |
| | 150000 | 5371 | 5478 | 5465 | 5609 | 77.6 | 67.1 | 97.8 | 60.9 | 5481 | 75.9 |
| ADAM | 5000 | 1672 | 1668 | 1707 | 1714 | 86.2 | 78.2 | 95.7 | 76.6 | 1690 | 84.2 |
| | 10000 | 3321 | 3348 | 3392 | 3408 | 86.2 | 81.7 | 95.7 | 80.8 | 3367 | 86.1 |
| | 150000 | 4989 | 5030 | 5067 | 5092 | 80.2 | 81.7 | 95.5 | 80.8 | 5045 | 84.6 |
| SAM | 5000 | 2661 | 2717 | 2729 | 2689 | 84.9 | 74 | 98.2 | 72.6 | 2699 | 82.4 |
| | 10000 | 5378 | 5425 | 5479 | 5392 | 87.1 | 75.7 | 98.1 | 73.2 | 5419 | 83.5 |
| | 150000 | 8113 | 8125 | 8228 | 8073 | 87.2 | 77.1 | 98.4 | 73.8 | 8135 | 84.1 |
| GENIE(OURS) | 5000 | 1862 | 2017 | 1532 | 1419 | 89.3 | 84.1 | 98.7 | 81.6 | 1708 | **88.4** |
| | 10000 | 3732 | 4054 | 3065 | 2836 | 87.6 | 80.9 | 98.4 | 81.6 | 3422 | 87.1 |
| | 150000 | 5607 | 6077 | 4586 | 4256 | 88.2 | 80.1 | 98.4 | 80.9 | 5132 | 86.9 |

*Table 15.* Comparison of Optimizers on VLCS Dataset Across Iterations.

| OPTIMIZER | ITERATION | TRAINING TIME (/S) | | | | ACCURACY | | | | AVG. | |
|---|---|---|---|---|---|---|---|---|---|---|---|
| | | [0] | [1] | [2] | [3] | [0] | [1] | [2] | [3] | TIME | ACC |
| SGD | 5000 | 9154 | 4947 | 9315 | 9119 | 97 | 61.9 | 73.3 | 74.6 | 8134 | 76.7 |
| | 10000 | 18311 | 10054 | 18641 | 18312 | 97.6 | 60.3 | 72.7 | 77.2 | 16330 | 77 |
| | 150000 | 27732 | 14956 | 27918 | 27419 | 98.1 | 62.4 | 72.6 | 77.8 | 24506 | 77.7 |
| ADAM | 5000 | 9087 | 4900 | 9154 | 9210 | 98.1 | 63 | 73 | 73.8 | 8088 | 77 |
| | 10000 | 18148 | 10024 | 18404 | 18312 | 98.1 | 63 | 73 | 73.8 | 16222 | 77 |
| | 150000 | 27532 | 14963 | 27615 | 27387 | 98.1 | 63 | 73 | 73.8 | 24374 | 77 |
| SAM | 5000 | 9544 | 5611 | 9523 | 9555 | 99 | 63.5 | 74.6 | 80.5 | 8558 | 79.4 |
| | 10000 | 19068 | 11203 | 18984 | 18749 | 98.9 | 64.1 | 76.1 | 81.9 | 17001 | 80.3 |
| | 150000 | 28510 | 16817 | 28772 | 27860 | 99 | 64.6 | 75.3 | 82.6 | 25490 | 80.4 |
| GENIE(OURS) | 5000 | 8726 | 4631 | 7118 | 5485 | 99.5 | 68.6 | 76.9 | 80.4 | 6490 | **81.4** |
| | 10000 | 17452 | 9273 | 14235 | 10885 | 99.5 | 68.6 | 76.9 | 80.4 | 12961 | **81.4** |
| | 150000 | 26164 | 13917 | 21396 | 16314 | 99.5 | 68.6 | 76.9 | 80.4 | 19448 | **81.4** |

*Table 16.* Comparison of Optimizers on OfficeHome Dataset Across Iterations.

| OPTIMIZER | ITERATION | TRAINING TIME (/S) | | | | ACCURACY | | | | AVG. | |
|---|---|---|---|---|---|---|---|---|---|---|---|
| | | [0] | [1] | [2] | [3] | [0] | [1] | [2] | [3] | TIME | ACC |
| SGD | 5000 | 6,634 | 6,436 | 5,842 | 4,554 | 46.2 | 40.1 | 56.8 | 61.9 | 5,867 | 51.3 |
| | 10000 | 13,334 | 12,665 | 11,722 | 8,899 | 56.8 | 46.4 | 70.6 | 76.2 | 11,655 | 62.5 |
| | 150000 | 20,031 | 18,565 | 17,676 | 13,175 | 58.5 | 47.1 | 73.4 | 76.6 | 17,362 | 63.9 |
| ADAM | 5000 | 8,000 | 8,147 | 5,799 | 4,255 | 57.8 | 49.4 | 73.8 | 73.4 | 6,550 | 63.6 |
| | 10000 | 16,260 | 16,602 | 11,605 | 8,379 | 59.8 | 52.2 | 73.8 | 74.8 | 13,212 | 65.2 |
| | 150000 | 24,776 | 25,466 | 17,387 | 13,062 | 59.8 | 52.2 | 73.8 | 74.8 | 20,173 | 65.2 |
| SAM | 5000 | 6,639 | 6,598 | 5,884 | 5,151 | 65.4 | 54.2 | 77.1 | 81.1 | 6,068 | 69.4 |
| | 10000 | 13,359 | 12,964 | 11,813 | 10,189 | 65.1 | 54.5 | 77.9 | 80.9 | 12,081 | 69.6 |
| | 150000 | 20,140 | 18,983 | 17,722 | 14,946 | 65.5 | 55.1 | 78.6 | 80.9 | 17,948 | **70** |
| GENIE(OURS) | 5000 | 4,777 | 5,846 | 4,025 | 4,061 | 66.5 | 55.4 | 77.8 | 80.3 | 4,677 | **70** |
| | 10000 | 9,553 | 11,624 | 8,062 | 8,211 | 65.8 | 53.3 | 77.8 | 79.9 | 9,363 | 69.2 |
| | 150000 | 14,261 | 17,333 | 12,229 | 12,369 | 65.8 | 53.3 | 77.1 | 80.3 | 14,048 | 69.1 |

## E.3. Integration with DG methods

The detailed performance of previous DG methods employed with our optimizer is presented in Table 17, Table 18, Table 19, Table 20.

*Table 17.* Integration with existing DG algorithms on the PACS dataset.

| ALGORITHM | ART | CARTOON | PHOTO | SKETCH | AVG. |
|---|---|---|---|---|---|
| ERM†(VAPNIK, 1999) | 84.7±0.4 | 80.8±0.6 | 97.2±0.3 | 79.3±1.0 | 85.5 |
| IRM†(ARJOVSKY ET AL., 2019) | 84.8±1.3 | 76.4±1.1 | 96.7±0.6 | 76.1±1.0 | 83.5 |
| GROUPDRO†(SAGAWA ET AL., 2020) | 83.5±0.9 | 79.1±0.6 | 96.7±0.3 | 78.3±2.0 | 84.4 |
| I-MIXUP†(XU ET AL., 2020) | 86.1±0.5 | 78.9±0.8 | 97.6±0.1 | 75.8±1.8 | 84.6 |
| MLDG†(LI ET AL., 2018A) | 85.5±1.4 | 80.1±1.7 | 97.4±0.3 | 76.6±1.1 | 84.9 |
| MMD†(LI ET AL., 2018B) | 86.1±1.4 | 79.4±0.9 | 96.6±0.2 | 76.5±0.5 | 84.7 |
| DANN†(GANIN ET AL., 2016) | 86.4±0.8 | 77.4±0.8 | 97.3±0.4 | 73.5±2.3 | 83.7 |
| CDANN†(LI ET AL., 2018C) | 84.6±1.8 | 75.5±0.9 | 96.8±0.3 | 73.5±0.6 | 82.6 |
| MTL†(BLANCHARD ET AL., 2021) | 87.5±0.8 | 77.1±0.5 | 96.4±0.8 | 77.3±1.8 | 84.6 |
| SAGNET†(NAM ET AL., 2021) | 87.4±1.0 | 80.7±0.6 | 97.1±0.1 | 80.0±0.4 | 86.3 |
| ARM†(ZHANG ET AL., 2021) | 86.8±0.6 | 76.8±0.5 | 97.4±0.3 | 79.3±1.2 | 85.1 |
| VREX†(KRUEGER ET AL., 2021) | 86.0±1.6 | 79.1±0.6 | 96.9±0.5 | 77.7±1.7 | 84.9 |
| MIXSTYLE(ZHOU ET AL., 2021) | 86.8±0.5 | 79.0±1.4 | 96.6±0.1 | 78.5±2.3 | 85.2 |
| RSC+GENIE(OURS) | 87.8±0.6 | 82.4±0.8 | 97.4±0.9 | 81.4±2.0 | 87.3 |
| CORAL+GENIE(OURS) | **88.8±0.2** | **82.5±0.5** | **98.2±0.1** | **82.2±0.2** | **87.9** |

*Table 18.* Integration with existing DG algorithms on the VLCS dataset.

| ALGORITHM | CALTECH | LABELME | SUN | VOC | AVG. |
|---|---|---|---|---|---|
| ERM†(VAPNIK, 1999) | 98.0±0.3 | 64.7±1.2 | 71.4±1.2 | 75.2±1.6 | 77.3 |
| IRM†(ARJOVSKY ET AL., 2019) | 98.6±0.1 | 64.9±0.9 | 73.4±0.6 | 77.3±0.9 | 78.6 |
| GROUPDRO†(SAGAWA ET AL., 2020) | 97.3±0.3 | 63.4±0.9 | 69.5±0.8 | 76.7±0.7 | 76.7 |
| I-MIXUP†(XU ET AL., 2020) | 98.3±0.6 | 64.8±1.0 | 72.1±0.5 | 74.3±0.8 | 77.4 |
| MLDG†(LI ET AL., 2018A) | 97.4±0.2 | 65.2±0.7 | 71.0±1.4 | 75.3±1.0 | 77.2 |
| MMD†(LI ET AL., 2018B) | 97.7±0.1 | 64.0±1.1 | 72.8±0.2 | 75.3±3.3 | 77.5 |
| DANN†(GANIN ET AL., 2016) | 99.0±0.3 | 65.1±1.4 | 73.1±0.3 | 77.2±0.6 | 78.6 |
| CDANN†(LI ET AL., 2018C) | 97.1±0.3 | 65.1±1.2 | 70.7±0.8 | 77.1±1.5 | 77.5 |
| MTL†(BLANCHARD ET AL., 2021) | 97.8±0.4 | 64.3±0.3 | 71.5±0.7 | 75.3±1.7 | 77.2 |
| SAGNET†(NAM ET AL., 2021) | 97.9±0.4 | 64.5±0.5 | 71.4±1.3 | 77.5±0.5 | 77.8 |
| ARM†(ZHANG ET AL., 2021) | 98.7±0.2 | 63.6±0.7 | 71.3±1.2 | 76.7±0.6 | 77.6 |
| VREX†(KRUEGER ET AL., 2021) | 98.4±0.3 | 64.4±1.4 | 74.1±0.4 | 76.2±1.3 | 78.3 |
| MIXSTYLE(ZHOU ET AL., 2021) | 98.6±0.3 | 64.5±1.1 | 72.6±0.5 | 75.7±1.7 | 77.9 |
| RSC+GENIE(OURS) | **99.1±0.3** | **68.3±1.3** | 76.0±0.4 | **79.1±0.6** | 80.6 |
| CORAL+GENIE(OURS) | 99.0±0.3 | **68.3±1.0** | **76.6±1.3** | 78.9±0.9 | **80.7** |

*Table 19.* Integration with existing DG algorithms on the OfficeHome dataset.

| ALGORITHM | ART | CLIPART | PRODUCT | REAL-WORLD | AVG. |
|---|---|---|---|---|---|
| ERM†(VAPNIK, 1999) | 61.3±0.7 | 52.4±0.3 | 75.8±0.1 | 76.6±0.3 | 66.5 |
| IRM†(ARJOVSKY ET AL., 2019) | 58.9±2.3 | 52.2±1.6 | 72.1±2.9 | 74.0±2.5 | 64.3 |
| GROUPDRO†(SAGAWA ET AL., 2020) | 60.4±0.7 | 52.7±1.0 | 75.0±0.7 | 76.0±0.7 | 66 |
| I-MIXUP†(XU ET AL., 2020) | 62.4±0.8 | 54.8±0.6 | 76.9±0.3 | 78.3±0.2 | 68.1 |
| MLDG†(LI ET AL., 2018A) | 61.5±0.9 | 53.2±0.6 | 75.0±1.2 | 77.5±0.4 | 66.8 |
| MMD†(LI ET AL., 2018B) | 60.4±0.2 | 53.3±0.3 | 74.3±0.1 | 77.4±0.6 | 66.4 |
| DANN†(GANIN ET AL., 2016) | 59.9±1.3 | 53.0±0.3 | 73.6±0.7 | 76.9±0.5 | 65.9 |
| CDANN†(LI ET AL., 2018C) | 61.5±1.4 | 50.4±2.4 | 74.4±0.9 | 76.6±0.8 | 65.7 |
| MTL†(BLANCHARD ET AL., 2021) | 61.5±0.7 | 52.4±0.6 | 74.9±0.4 | 76.8±0.4 | 66.4 |
| SAGNET†(NAM ET AL., 2021) | 63.4±0.2 | 54.8±0.4 | 75.8±0.4 | 78.3±0.3 | 68.1 |
| ARM†(ZHANG ET AL., 2021) | 58.9±0.8 | 51.0±0.5 | 74.1±0.1 | 75.2±0.3 | 64.8 |
| VREX†(KRUEGER ET AL., 2021) | 60.7±0.9 | 53.0±0.9 | 75.3±0.1 | 76.6±0.5 | 66.4 |
| MIXSTYLE(ZHOU ET AL., 2021) | 51.1±0.3 | 53.2±0.4 | 68.2±0.7 | 69.2±0.6 | 60.4 |
| RSC+GENIE(OURS) | 63.2±2.5 | 54.8±0.3 | 76.0±0.7 | 78.3±1.9 | 68.1 |
| CORAL+GENIE(OURS) | **66.5±0.2** | **56.7±0.3** | **78.8±0.1** | **80.4±0.6** | **70.6** |

*Table 20.* Integration with existing DG algorithms on the TerraIncognita dataset.

| ALGORITHM | L100 | L38 | L43 | L46 | AVG. |
|---|---|---|---|---|---|
| ERM†(VAPNIK, 1999) | 54.3±0.4 | 42.5±0.7 | 55.6±0.3 | 38.8±2.5 | 47.8 |
| IRM†(ARJOVSKY ET AL., 2019) | 54.6±1.3 | 39.8±1.9 | 56.2±1.8 | 39.6±0.8 | 47.6 |
| GROUPDRO†(SAGAWA ET AL., 2020) | 41.2±0.7 | 38.6±2.1 | 56.7±0.9 | 36.4±2.1 | 43.2 |
| I-MIXUP†(XU ET AL., 2020) | **59.6±2.0** | 42.2±1.4 | 55.9±0.8 | 33.9±1.4 | 47.9 |
| MLDG†(LI ET AL., 2018A) | 54.2±3.0 | 44.3±1.1 | 55.6±0.3 | 36.9±2.2 | 47.8 |
| MMD†(LI ET AL., 2018B) | 41.9±3.0 | 34.8±1.0 | 57.0±1.9 | 35.2±1.8 | 42.2 |
| DANN†(GANIN ET AL., 2016) | 51.1±3.5 | 40.6±0.6 | 57.4±0.5 | 37.7±1.8 | 46.7 |
| CDANN†(LI ET AL., 2018C) | 47.0±1.9 | 41.3±4.8 | 54.9±1.7 | 39.8±2.3 | 45.8 |
| MTL†(BLANCHARD ET AL., 2021) | 49.3±1.2 | 39.6±6.3 | 55.6±1.1 | 37.8±0.8 | 45.6 |
| SAGNET†(NAM ET AL., 2021) | 53.0±2.9 | 43.0±2.5 | **57.9±0.6** | 40.4±1.3 | 48.6 |
| ARM†(ZHANG ET AL., 2021) | 49.3±0.7 | 38.3±2.4 | 55.8±0.8 | 38.7±1.3 | 45.5 |
| VREX†(KRUEGER ET AL., 2021) | 48.2±4.3 | 41.7±1.3 | 56.8±0.8 | 38.7±3.1 | 46.4 |
| MIXSTYLE(ZHOU ET AL., 2021) | 54.3±1.1 | 34.1±1.1 | 55.9±1.1 | 31.7±2.1 | 44 |
| RSC+GENIE(OURS) | 56.5±3.2 | **44.5±3.7** | 55.9±1.0 | **40.9±0.3** | **49.5** |
| CORAL+GENIE(OURS) | 57.0±1.2 | 42.9±0.7 | 54.1±0.8 | 39.4±1.3 | 48.4 |

### E.4. Detailed SDG performance.

This section reports a detailed comparison of our optimizer with existing optimizers across four datasets in the SDG setting. It also provides the performance of previous methods when employed with our optimizer. Table 21, Table 22 Table 23, Table 24

### E.5. Model Analysis

Figure 6. visualizes the behavior of SGD, Adam, and GENIE on various loss landscapes, with the number of iterations fixed at 30 for consistency. The results show that GENIE does not directly converge to the minima of the training data, whereas Adam and SGD quickly converge to the minima. While this behavior might be advantageous in IID scenarios, it can lead to overfitting to the source domain in OOD settings.Figure 7 provides a more detailed analysis of the experiment. It further elaborates on the parameter update patterns observed across optimizer.

*Table 21.* Detailed performance on the PACS dataset in the SDG setting.

| ALGORITHM | ART | CARTOON | PHOTO | SKETCH | AVG. |
|---|---|---|---|---|---|
| ERM+ADAM | 77.5 | 72.1 | **54.4** | 53.3 | 64.3 |
| ERM+SGD | 64.0 | 66.0 | 40.8 | 27.1 | 49.5 |
| ERM+SAM | 70.1 | 75.6 | 42.5 | 42.7 | 57.7 |
| ERM+GENIE(OURS) | **78.6±0.6** | **81.9±0.9** | 53.8±2.2 | **63.5±5.2** | **69.5** |
| RSC+GENIE(OURS) | 79.8±2.0 | 81.0±1.2 | 56.7±3.3 | 65.9±4.6 | 70.9 |
| CORAL+GENIE(OURS) | 78.9±0.6 | 78.6±2.2 | 53.9±1.2 | 61.3±3.4 | 68.2 |

*Table 22.* Detailed performance on the VLCS dataset in the SDG setting.

| ALGORITHM | CALTECH | LABELME | SUN | VOC | AVG. |
|---|---|---|---|---|---|
| ERM+ADAM | 33.7 | 53.5 | 62.3 | 75.4 | 56.2 |
| ERM+SGD | 46.3 | 52.4 | 65.6 | 77.3 | 60.4 |
| ERM+SAM | 54.4 | 68.1 | 65.8 | 78.7 | 66.7 |
| ERM+GENIE(OURS) | **56.6±4.3** | **75.4±1.1** | **67.2±1.4** | **80.5±0.3** | **69.9** |
| RSC+GENIE(OURS) | 56.3±2.5 | 72.0±1.5 | 68.8±1.7 | 79.6±1.1 | 69.2 |
| CORAL+GENIE(OURS) | 55.9±1.3 | 71.7±1.8 | 67.2±1.8 | 79.9±1.4 | 68.7 |

*Table 23.* Detailed performance on the OfficeHome dataset in the SDG setting.

| ALGORITHM | ART | CLIPART | PRODUCT | REAL-WORLD | AVG. |
|---|---|---|---|---|---|
| ERM+ADAM | 52.6 | 46.5 | 45.6 | 58 | 50.7 |
| ERM+SGD | 47.7 | 41.4 | 43.2 | 51.5 | 45.9 |
| ERM+SAM | **60.1** | 57.8 | **55.8** | **63.1** | **59.2** |
| ERM+GENIE(OURS) | 59.4 ± 0.7 | **58.7 ± 0.9** | 54.1 ± 0.5 | 62.0 ± 0.3 | 58.6 |
| RSC+GENIE(OURS) | 55.8 ± 2.6 | 52.5 ± 2.0 | 49.7 ± 3.0 | 59.7 ± 2.0 | 54.4 |
| CORAL+GENIE(OURS) | 58.0 ± 0.9 | 54.8 ± 1.5 | 51.3 ± 0.3 | 61.4 ± 0.7 | 56.4 |

*Table 24.* Detailed performance on the TerraIncognita dataset in the SDG setting.

| ALGORITHM | L100 | L38 | L43 | L46 | AVG. |
|---|---|---|---|---|---|
| ERM+ADAM | 27.0 | 25.5 | **42.9** | 38.6 | 33.5 |
| ERM+SGD | 22.1 | 17.9 | 22.3 | 29 | 22.8 |
| ERM+SAM | 21.1 | 21.6 | 30.3 | 34.2 | 26.8 |
| ERM+GENIE(OURS) | **28.5 ± 0.5** | **29.4 ± 1.5** | 41.7 ± 1.8 | **44.5 ± 1.3** | **36.0** |
| RSC+GENIE(OURS) | 27.1 ± 1.5 | 22.6 ± 1.7 | 39.7 ± 3.0 | 43.3 ± 1.1 | 33.2 |
| CORAL+GENIE(OURS) | 29.3 ± 1.0 | 29.8 ± 1.8 | 43.9 ± 1.3 | 43.6 ± 1.6 | 36.7 |

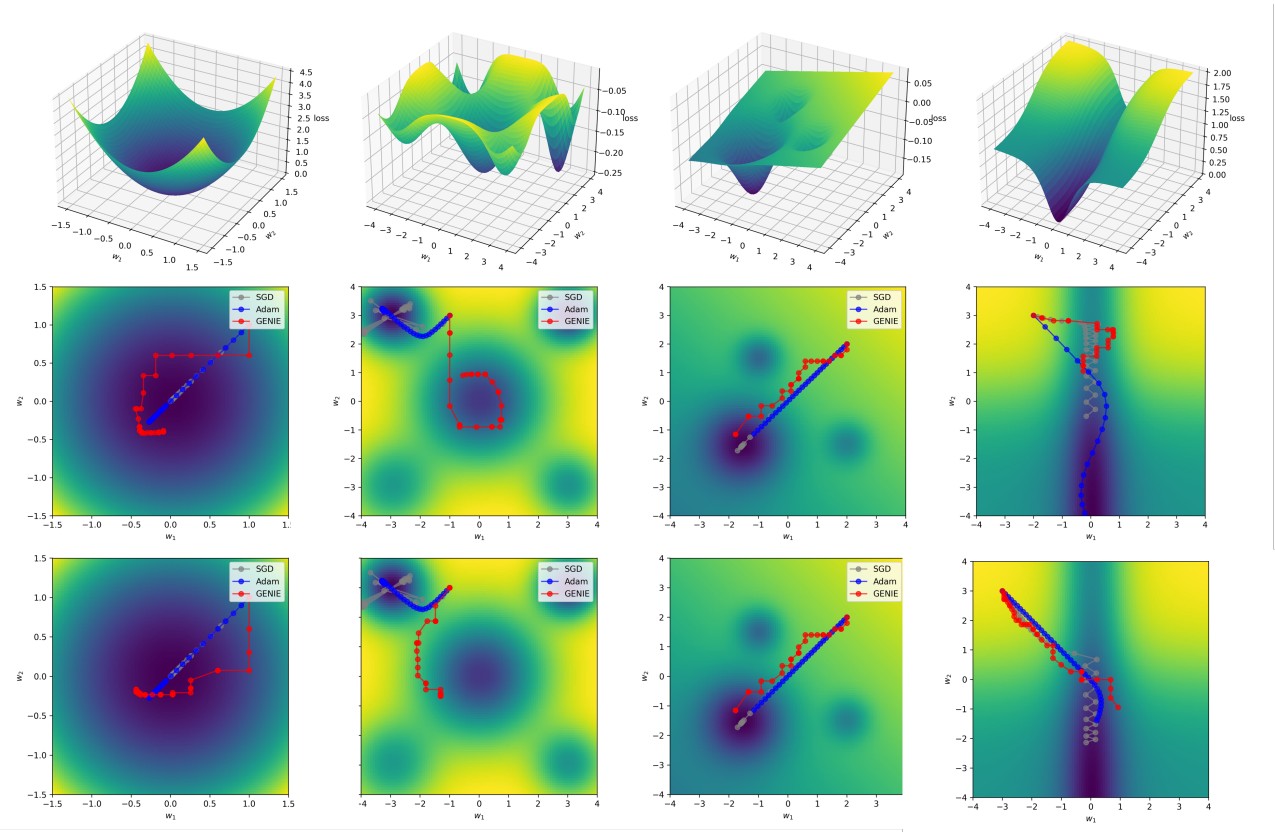

*Figure 6.* Optimization trajectories on a simulated loss landscape.

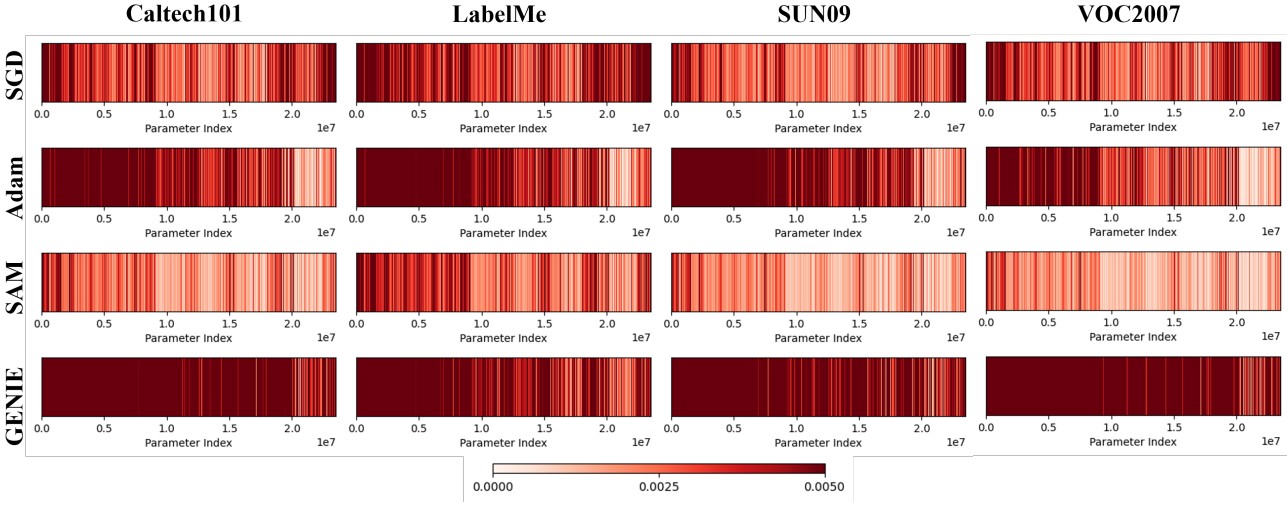

*Figure 7.* Heatmaps visualizing normalized parameter update magnitudes by parameter ID for different optimizers throughout training on the VLCS dataset in the DG.

