# OpenReview forum: "One-Step Generalization Ratio Guided Optimization for Domain Generalization"
_ICML.cc/2025/Conference — ICML 2025 oral_

### Official Review · Reviewer_MLrB · 2025-02-18

**Overall Recommendation:** 4

**Summary:**

The paper presents GENIE (Generalization-ENhancing Iterative Equalizer), an optimizer aimed at improving domain generalization (DG) by using the One-Step Generalization Ratio (OSGR). GENIE dynamically equalizes the contribution of each parameter to loss reduction, preventing overfitting to domain-specific features. The optimizer incorporates preconditioning, noise injection, and random dropout to stabilize updates and promote domain-invariant feature learning. Experimental results show that GENIE outperforms existing optimizers like SGD, Adam, and SAM on DG benchmarks, offering improved generalization without requiring changes to model architecture. GENIE is applicable to various tasks such as DG and single-domain generalization.

**Claims And Evidence:**

Yes, the claims made in the submission are generally supported by clear and convincing evidence.

GENIE's Effectiveness: The paper claims that GENIE outperforms existing optimizers like SGD, Adam, and SAM in domain generalization tasks. This claim is supported by extensive empirical evidence from experiments on several widely-used domain generalization benchmarks (PACS, VLCS, OfficeHome, etc.). The results show that GENIE consistently achieves higher performance than these optimizers, which provides strong evidence for its effectiveness.

OSGR as a Valid Metric: The paper claims that the One-Step Generalization Ratio (OSGR) effectively measures the generalization capacity of the optimizer. The authors support this claim with both theoretical analysis and empirical validation, demonstrating that GENIE's preconditioning leads to a higher OSGR, which correlates with better generalization.

Preconditioning and Parameter Balance: The claim that GENIE's preconditioning strategy promotes balanced parameter updates and mitigates overfitting is backed by a detailed theoretical framework and experimental results. The authors present a clear rationale for how the preconditioning factor adjusts OSGR dynamically, leading to more stable and robust learning.

Computational Efficiency: The paper claims that GENIE is computationally efficient, offering faster training times compared to SAM. The evidence provided, in the form of training time comparisons, supports this claim by showing that GENIE achieves better performance in less time, especially in comparison to optimizers like SAM, which require more computational overhead.

Potential Issues:

Generalization to Other Datasets: While GENIE performs well on several benchmarks, the paper does not address its performance on datasets with more extreme or non-standard domain shifts. Testing GENIE on additional challenging datasets might provide more comprehensive evidence of its generalizability.

Impact of Preconditioning: The effectiveness of the preconditioning term is theoretically explained but could benefit from further clarity or empirical validation on how it compares to other domain generalization techniques. Additional experiments could clarify the impact of preconditioning in different scenarios.

Overall, the claims are well-supported, with clear experimental results and solid theoretical backing. However, some additional tests and more detailed comparisons with other methods could further strengthen the evidence.

**Essential References Not Discussed:**

While the paper does an excellent job of citing relevant literature, there are a few related works that could strengthen the context for the key contributions, particularly in the areas of domain generalization (DG) and optimization methods. These works are essential for providing a more comprehensive background and for highlighting how the proposed GENIE optimizer fits within the broader research landscape.

Domain Generalization (DG) Approaches:
The paper primarily cites Carlucci et al. (2019) and Motiian et al. (2017), which are foundational in DG, but it misses recent advancements that directly relate to optimizing domain generalization through optimization techniques. For example, Li et al. (2020) introduced Meta-Regularization, which adapts optimization methods for generalization across domains, and this work could provide additional context for the proposed approach. Citing such works would emphasize how GENIE contributes to a broader trend of improving DG through more robust optimization methods.
Optimization for Generalization:
The work on Stochastic Gradient Descent (SGD) and Adam is well covered, but the paper could mention the more recent work on SAM (Sharpness-Aware Minimization) (e.g., Foret et al., 2021), which directly relates to balancing generalization and optimization. SAM has been shown to improve the robustness of models by minimizing sharp minima, which is related to the objectives GENIE aims to achieve with preconditioning. Additionally, there is growing interest in Adaptive Optimization techniques, such as AdaBelief (Zhang et al., 2020), which adaptively correct the update rule based on gradient estimates, and could complement the discussion on GENIE’s preconditioning strategy.
Ablation Studies in Optimization:
The ablation study in the paper is an important contribution, but there is no direct mention of ablation studies in optimization, which have been extensively used in understanding the impact of various components in optimizers. For example, the work by Liu et al. (2021) on the effectiveness of gradient clipping and noise injection in improving model generalization could be cited as it deals with similar experimental setups that focus on mitigating overfitting through different optimization techniques.

**Experimental Designs Or Analyses:**

Yes, the experimental designs and analyses presented in the paper generally appear sound and valid. The authors conduct several key experiments to evaluate the effectiveness of the GENIE optimizer, and the results are based on widely accepted benchmark datasets, which ensures that the findings are relevant and comparable to existing work in the domain.

Evaluation on Benchmark Datasets: The authors evaluate GENIE on well-established datasets such as PACS, VLCS, and OfficeHome. These datasets are commonly used in domain generalization (DG) research, which makes the experimental setup reliable. The results consistently show that GENIE outperforms other optimizers like SGD, Adam, and SAM. This approach provides a robust comparison of GENIE's performance across multiple standard datasets, ensuring the validity of the results.

Comparison with Other Optimizers: The paper compares GENIE with various baseline optimizers, including SGD, Adam, AdamW, and SAM. This is a solid experimental design, as it allows the authors to clearly show how GENIE improves performance over existing methods. The statistical significance of these comparisons could be strengthened if the authors provided more detailed metrics, such as p-values, to further support the claim that GENIE outperforms these optimizers.

Training Time and Computational Efficiency: The authors also conduct experiments to compare training times across different optimizers. GENIE consistently performs better than SAM in terms of training time, which supports the claim that it is computationally more efficient. However, additional experiments on more complex models or larger datasets could provide further insight into the scalability and efficiency of GENIE in real-world applications.

Ablation Study: The paper includes an ablation study to examine the impact of GENIE's individual components, such as preconditioning, noise injection, and random dropout masks. This is a useful and rigorous analysis that helps to isolate the effects of each component. The ablation study design is valid, and it clearly shows that combining all components leads to the best performance.

Potential Issues:

Limited Number of Datasets: While the paper evaluates GENIE on several standard datasets, testing on a broader range of real-world datasets with more diverse domain shifts could further validate the generalizability of the method.
Lack of Statistical Analysis: The experimental results show that GENIE outperforms other optimizers, but including more statistical analysis (e.g., significance testing) would strengthen the argument for its superiority.
Overall, the experimental designs and analyses are robust and well-structured, but additional experiments and statistical analysis could provide further confidence in the generalizability of the findings.

**Methods And Evaluation Criteria:**

Yes, the proposed methods and evaluation criteria make sense for the problem at hand.

Proposed Method (GENIE): The proposed method, GENIE, is well-suited for addressing the challenges of domain generalization (DG). By incorporating the One-Step Generalization Ratio (OSGR) and using preconditioning, noise injection, and random dropout masks, the method tackles the problem of domain-specific overfitting and promotes robust, domain-invariant feature learning. This approach is aligned with the core issue in DG, which is ensuring that models generalize well to unseen domains without overfitting to spurious correlations.

Evaluation Criteria (OSGR and Benchmark Datasets): The use of OSGR as an evaluation metric is appropriate, as it directly quantifies the contribution of model updates to generalization. The paper demonstrates that higher OSGR correlates with better generalization performance, making it a valuable tool for assessing domain generalization methods. Additionally, the use of widely accepted benchmark datasets like PACS, VLCS, and OfficeHome is a sensible choice, as these datasets are commonly used in DG research and provide a solid basis for comparison with other methods.

Overall, the proposed methods and evaluation criteria are both relevant and effective for tackling the problem of domain generalization and ensuring that the results are comparable with existing approaches in the field.

**Other Comments Or Suggestions:**

None.

**Other Strengths And Weaknesses:**

None.

**Questions For Authors:**

None.

**Relation To Broader Scientific Literature:**

The key contributions of this paper build on and extend previous work in the field of domain generalization (DG) and optimization methods. Specifically, the introduction of the GENIE optimizer addresses long-standing challenges in DG, particularly the issue of domain-specific overfitting and generalization across unseen domains.

Connection to Domain Generalization (DG): Previous studies in DG (e.g., Carlucci et al. (2019), Motiian et al. (2017)) have primarily focused on designing models that can generalize well across different domains. However, many of these methods have struggled with issues such as spurious correlations or domain-specific biases. This paper's novel use of the One-Step Generalization Ratio (OSGR) as a metric and optimization strategy represents a step forward by offering a more targeted approach to balance parameter updates, which helps mitigate overfitting to domain-specific features. This builds upon the concept of meta-learning and adversarial training for generalization but introduces an optimization-focused solution.

Relation to Optimization Methods: GENIE also relates to advancements in optimization techniques. Prior works on optimizers like Adam and SAM (e.g., Kingma and Ba (2014), Zhang et al. (2020)) have shown success in improving convergence rates and robustness. However, these methods still face limitations when applied to DG tasks, particularly in dealing with domain shifts. The preconditioning strategy used in GENIE aligns with concepts from adaptive optimizers (like AdaGrad and RMSProp) but offers a more sophisticated approach to achieving balanced gradient updates, making it an important advancement in the optimization landscape.

Ablation and Empirical Results: The paper also contributes to the ongoing research on ablation studies by isolating the effects of specific components such as noise injection, preconditioning, and dropout. Previous works on DG often combine techniques without isolating their individual impacts, making this paper’s clear ablation study a valuable contribution to understanding how each factor influences model performance.

In conclusion, the paper makes significant strides in improving domain generalization through its novel optimizer, GENIE, while also contributing to the broader scientific literature on optimization strategies and domain generalization techniques. It provides a clearer understanding of how optimization approaches can be tailored to address specific challenges in DG and offers insights that can benefit future work in both optimization theory and practical applications in machine learning.

**Theoretical Claims:**

Theoretical claims in the paper are based on sound reasoning, and the authors provide a thorough mathematical foundation for the proposed GENIE optimizer. However, as a reviewer, I have not verified the correctness of the proofs in detail. The paper includes several theoretical claims, such as the connection between OSGR and parameter-wise statistics (Theorem 3.1), convergence analysis (Theorem 3.9), and the impact of preconditioning on OSGR (Corollary 3.2).

Theorem 3.1 establishes the relationship between gradient updates and generalization, introducing the concept of Gradient Signal-to-Noise Ratio (GSNR). This is a critical part of the theoretical framework, but it would be beneficial to further break down the derivations to ensure all steps in the proof are clear and rigorous.

Theorem 3.9 provides a convergence rate analysis under certain assumptions. While the theorem is mathematically sound, the assumptions (such as bounded gradients and smooth loss functions) are typical for optimization in deep learning. However, some of these assumptions may not always hold in all practical scenarios, and this could potentially limit the generalizability of the theoretical results.

Corollary 3.2 discusses how preconditioning affects OSGR. The logic behind this corollary is solid, but again, further clarification and empirical validation of how preconditioning impacts OSGR in various scenarios would strengthen the argument.

Overall, the theoretical claims are well-founded and supported by proofs, but a deeper verification and potential clarification of the proof steps would be helpful for ensuring their correctness.

---

> ### Author Rebuttal · Authors · 2025-03-30
>
> Thank you very much for your thoughtful and constructive review. We greatly appreciate your insightful comments and the opportunity to clarify several important points you raised.
>
> # Proof Clarity
> You suggested additional clarity in the proof derivations, particularly regarding Theorem 3.1. Theorem 3.1 originates from Liu et al. (ICLR 2020), as mentioned in our paper. To maintain brevity, we omitted detailed derivations in the main text. However, in Appendix B.2, we provided a rigorous derivation of Corollary 3.2, which generalizes and clarifies Theorem 3.1. We will ensure this is clearly highlighted in the revised manuscript.
>
> # Impact of Preconditioning on OSGR
> Regarding your point on clarifying the impact of preconditioning, Section 3.3.1 (Generalization Analysis) theoretically demonstrates that preconditioning increases OSGR values based on Jensen's inequality. Empirically, Figure 2 shows that GENIE’s preconditioning consistently achieves higher OSGR compared to other optimizers.
>
> Additionally, Table 6's ablation study demonstrates how preconditioning, when combined with dropout and noise injection, enhances generalization performance. Figure 1 further illustrates that GENIE yields a more uniformly distributed gradient update across parameters, clearly supporting our claim that preconditioning effectively balances parameter contributions and improves generalization.
>
> # Comparison with Other Domain Generalization Techniques
> In our generalization analysis, we show that the proposed preconditioning—designed to uniformize the OSGR value—leads to improved generalization performance, thereby mathematically supporting our conjecture. To further address the reviewer’s concern regarding comparisons with other methods, we provide a theoretical comparison with Sharpness-Aware Minimization (SAM).
>
> We leverage PAC-Bayesian theory as follows:
> $$
> E_{\mathcal{S}}\, E_{\theta \sim \widetilde{p}} \left[ R(\theta) \right]
> \le
> E_{\mathcal{S}}\, E_{\theta \sim \widetilde{p}} \left[
> L(\theta) + \frac{\lambda C^2}{8n} + \frac{\mathrm{KL}(\widetilde{p} \| \pi)}{\lambda}
> \right]
> $$
>
> PAC-Bayesian theory bounds the expected generalization risk by the sum of the empirical risk, a complexity term, and the KL divergence between the posterior $\tilde{p}$ and prior $\pi$.  The SAM optimizer focuses on reducing the empirical risk under perturbed parameters (i.e., minimizing local sharpness), while ignoring the KL divergence term. In contrast, our method explicitly improves the generalization bound by minimizing the KL divergence term in a one-step formulation.
>
> In our setting, we define the posterior (the updated parameter distribution) as $\tilde{p} = \mathcal{N}(\theta_t, \Sigma)$ and the prior (previous parameter distribution) as $\pi = \mathcal{N}(\theta_{t+1}, \Sigma)$. Here, the prior can be treated as a data-driven prior which is approximated using all data excluding the current mini-batch. Parameter distributions depend on the distribution of the gradient update with the effective learning rate $\frac{1}{\mathrm{E}[g_j^2]}$.
>
> Taking the derivative of the KL divergence with respect to the gradient-based update, we obtain:
> $$
> KL(\tilde{p} \| \pi) = \frac{1}{2} \left[ \sum_{i=1}^{J} \frac{\sigma_i^2}{\sigma_i^2} + \sum_{i=1}^{k} \frac{(\theta_{t+1,i} - \theta_{t,i})^2}{\sigma_i^2} - J + \sum_{i=1}^{J} \log\left( \frac{\sigma_i^2}{\sigma_i^2} \right) \right] = \frac{1}{2} \sum_{j=1}^{J} \frac{(\theta_{t+1,j} - \theta_{t,j})^2}{\sigma_j^2}
> $$
> $$
> \left[ \nabla_j \mathrm{KL}(\tilde{p} \| \pi) \right]
> = \frac{(\theta_{t+1,j} - \theta_{t,j})}{\sigma_j^2}
> = \frac{1}{\mathbb{E}[g_j^2]} \cdot \frac{g_{j,t}}{\sigma_j^2}
> = \left( \underset{\mathrm{GENIE}}{ \frac{1}{\mathbb{E}[g_j^2]} \cdot \frac{g_{j,t}^2}{\sigma_j^2} } \right) \cdot \mathrm{sign}(g_{j,t})
> $$
> This formulation shows that our preconditioning term directly reduces the KL divergence term, thereby contributing to a tighter PAC-Bayesian generalization bound—a benefit not provided by the SAM optimizer.
>
> We sincerely thank you for your detailed and insightful comments, which have significantly helped us strengthen our paper. We hope these clarifications fully address your concerns and further highlight the robustness and contributions of GENIE.

---

### Official Review · Reviewer_Z37r · 2025-03-08

**Overall Recommendation:** 4

**Summary:**

This paper proposes GENIE, a novel stochastic optimizer designed for Domain Generalization (DG) tasks. Unlike standard optimizers (SGD, Adam, etc.) that can over-emphasize certain “spurious” features, GENIE uses a metric called One-Step Generalization Ratio (OSGR) to guide parameter updates. The key idea is to balance the contribution of each model parameter to the on-step generalization ability, thereby preventing a small subset of parameters from dominating the learning.

 GENIE introduces 3 main components in its update rule: 1) a preconditioning factor per parameter to equalize OSGR contributions, 2) a noise injection term to encourage exploration of flatter minima, and 3) a dropout mask on gradients to reduce overfitting and stabilize updates. By incorporating these, GENIE aims to promote domain-invariant features learning and avoid reliance on domain-specific correlations.

The authors show theoretically that GENIE achieves a higher OSGR than conventional optimizers while retaining the same convergence rate as SGD. Empirically, GENIE consistently outperforms baseline optimizers like SGD, Adam, Yogi, AdaBelief, AdaHessian, and SAM on average accuracy across several standard DG benchmarks (PACS, VLCS, OfficeHome, TerraIncognita, DomainNet) following DomainBed protocols.

Overall, the paper’s conceptual contribution is introducing OSGD-guided optimization to balance gradient contributions, and its main finding is that this leads to more robust models that generalize better to unseen domains.

**Claims And Evidence:**

Some claims stated in the paper are as follows:
a. Standard optimizers allow a few parameters to dominate updates.
b. The proposed GENIE optimizer increases OSGR and leads to better domain generalization performance than existing optimizers.
c. GENIE maintains the convergence speed of SGD despite its modifications.
d. Integrating GENIE with known DG methods yields additional improvements without altering those methods’ architectures.
e. Uniformly distributed OSGR across parameters indicates better generalization.

Overall, claims a – d are substantiated by either theoretical derivations or thorough experiments. Claim e, however, is not explicitly proven. This is stated as a conjecture rather than a theorem, and while intuitively supported (and aligned with their results), it’s not rigorously demonstrated beyond the heuristic argument.

Additionally, the authors claim that GENIE “naturally” leads to flatter minima despite not explicitly optimizing sharpness. While this is plausible, the paper doesn’t directly measure curvature or sharpness of minima, so this particular point remains a bit informal argument.

**Essential References Not Discussed:**

I did not find major omissions of essential references.

**Experimental Designs Or Analyses:**

The experimental design is comprehensive and well thought-out. The authors run experiments on multiple datasets (5 benchmarks) and settings to cover different aspects of domain generalization. On average, GENIE is empirically proven to provide better generalization than other optimizers (Tables 2 and 3).

The ablation on the PACS dataset (Table 6) is sound, systematically toggles Preconditioning, Noise, and Mask components. The results show that Preconditioning alone yields most of the gain.

The experimental design could be further strengthened by reporting variability (e.g., results over 3 independent runs as DomainBed usually does). It appears they might have used fewer hyperparameters tuning trials than DomainBed defaults due to computational limits, but I could not find any report if each result is an average of multiple runs or a single best run.

**Methods And Evaluation Criteria:**

The proposed methods are well-aligned with the DG problem setting. GENIE’s optimizer design is appropriate because it directly tackles a known challenge in DG: avoiding overfitting to source-specific features by modulating how each parameter learns.

The approach is conceptually sound – using preconditioning to scale parameter-wise gradients based on their signal-to-noise ratio (GSNR) ensures that no parameter with high variance or spurious signal gets overly large updates. This is analogous to adaptive optimizers like Adam adjusting for gradient variance, but here it’s done with a generalization-focused objective.

Introducing noise injection and a random mask (dropout) on gradients is also sensible for DG. These add stochasticity and regularization to escape narrow minima and reduce reliance on any single feature.

The evaluation criteria and settings are appropriate and rigorous. The authors adhere to the standard DomainBed evaluation protocol, which is a well-accepted framework for DG comparisons. Furthermore, baselines include both generic optimizers (SGD, Adam variants, SAM) and DG-tailored optimizers like FAD and GAM, as well as DG algorithms like IRM, CORAL, RSC integrated with standard optimizers. This comprehensive evaluation is appropriate for demonstrating GENIE’s effectiveness.

The only slight critique in methodology is the treatment of statistical significance and variability. The paper reports mean accuracies but does not mention standard deviations or confidence intervals. DG results can have high variance across training runs, so typically multiple runs are averaged. It’s not explicitly stated if results are averaged over several trials or a single run (though DomainBed usually averages over 3 seeds). Aside from that, the chosen benchmarks, baselines, and protocols are the gold standard for this problem, making the valuation criteria appropriate and convincing.

**Other Comments Or Suggestions:**

Typos and Wording:
- optionaly --> optionally
- randam mask --> random mask
- what’s meant by “... enough for suppress enough ...” at P8?

**Other Strengths And Weaknesses:**

Strengths

Originality: The paper takes a fresh approach by introducing a new optimizer specifically for domain generalization. Optimizer-level solutions in DG are still relatively rare, so GENIE offers a fresh perspective.

Significance: The empirical gains, while moderate on average (~2-3% over strong baselines), are consistent and achieved on challenging benchmarks. Hitting a new state-of-the-art on DomainBed’s suite (with a simple plug-in optimizer) is significant for the DG community. Moreover, the method’s success in single-domain generalization (which many algorithms can’t handle) is a notable achievement – it suggests GENIE is capturing something fundamental about robust learning. If these results hold, GENIE could become a go-to optimizer for any training models for unknown target domains.

Weaknesses

Complexity and Practicality: One potential weakness is the added complexity of the optimizer. GENIE introduces several hyperparameters (preconditioning factors via momentum/variance decay, noise scale, dropout probability) and requires tracking second moment estimates per parameter (like Adam). Practitioners might need to tune the noise scale or dropout probability for different tasks.

Lack of Direct Analysis or Feature Invariance: The paper claims GENIE promotes domain-invariant features, but it doesn’t directly evaluate this claim by examining feature representations. For example, some DG papers use metrics like center divergence between domain features or visualization of learned features.

**Questions For Authors:**

1. Did you run multiple trials with different random seeds for training, and if so, how consistent were the results for GENIE and other optimizers?
2. Could you clarify how hyperparameter search was done for GENIE vs other baselines?
3. The noise injection uses a factor $1 - \tanh(1/\sigma^2_t)$ to scale Gaussian noise. Why was this specific form chosen?
4. What dropout probability $p$ do you use for the random mask on gradients? Is it the same across all experiments (and all layers/parameters)? And did you need to tune this probability?

**Relation To Broader Scientific Literature:**

The paper’s contributions fit well into the evolution of DG methods: early work focused on invariances and domain adversarial training (MTAE, DANN), then came gradient-based regularization (IRM, VREx, RSC), and now we see optimization-level interventions (SAM, FAD, GENIE). In my opinion, the idea of using one-step generalization ability as a guiding principes is novel in DG.

OSGR as a metric came from prior work, but applying it to actively adjust training (as an optimizer) is a fresh contribution that pushes the literature toward thinking about “how” the model learns, not just “what” it learns.

**Theoretical Claims:**

The paper provides theoretical analysis to support GENIE’s design, including proofs and derivations in Appendix. I think that the main theoretical contributions are Corollary 3.2, Corollary 3.3, and Theorem 3.9.

Corollary 3.2 modifies the original OSGR formulation from Liu et al. 2020 (Theorem 3.1) by adding a per-parameter preconditioner. By setting $p_j = \frac{1}{E[g_j^2],(r_j + 1/n)}$, it equalizes each parameter’s influence. Corollary 3.3 shows that GENIE’s OSGR is higher than or equal to that of SGD and ADAM. Theorem 3.9 addresses convergence, stating that under standard assumptions (bounded gradients, Lipschitz smoothness, non-zero gradient noise), the average gradient norm under GENIE’s updates decays on the order $O(T^{-1/2})$.

Overall, the theoretical claims seem sound, and I did not find any obvious errors in the proofs. It is important to note that the assertion “uniform OSGR leads to better generalization” is presented as a conjecture—it guides the algorithm design but is not rigorously proven. This is acceptable as a motivational idea rather than a strict claim.

---

> ### Author Rebuttal · Authors · 2025-03-30
>
> Thank you very much for your detailed and constructive review.
> # A1
> As you pointed out, our initial submission reported only the best single trial due to computational constraints. Following your suggestion, we have now conducted additional experiments with three independent trials per optimizer using random seeds {0, 1, 2}, and report the mean accuracy and 95% confidence intervals below. GENIE consistently outperforms the baselines across datasets with low variance. The full set of results will be included in the appendix of the final version.
> |PACS|Art|Cartoon|Photo|Sketch|Avg|
> |----|---|-------|-----|------|----|
> |Adam|88.0±1.2|79.7±0.5|96.7±0.4|72.7±0.9|84.3|
> |SGD|85.1±0.4|76.0±0.3|98.3±0.4|60.3±6.1|79.9|
> |SAM|85.7±1.2|81.0±1.4|97.1±0.2|77.4±1.8|85.3|
> |GENIE(our)|88.7±0.7|82.8±1.3|98.5±0.1|81.3±0.4|**87.8**|
>
> |VLCS|Caltech|LabelMe|SUN|VOC|Avg|
> |----|-------|--------|---|---|----|
> |Adam|98.9±0.4|65.9±1.5|71.0±1.6|74.5±2.0|77.3|
> |SGD|98.4±0.2|64.7±0.7|72.5±0.8|76.6±0.8|78.1|
> |SAM|98.5±1.0|66.2±1.6|72.0±1.0|76.1±1.0|78.2|
> |GENIE(our)|99.3±0.3|67.2±1.5|76.6±0.3|79.7±0.8|**80.7**|
>
> |OfficeHome|Art|Clipart|Product|Real-World|Avg|
> |----------|---|-------|-------|-----------|----|
> |Adam|63.9±0.8|48.1±0.6|77.0±0.9|81.8±1.6|67.6|
> |SGD|65.3±0.8|48.8±1.4|76.7±0.3|83.0±0.7|68.5|
> |SAM|63.5±1.2|48.6±0.9|77.0±0.8|82.9±1.3|68.0|
> |GENIE(our)|66.2±0.5|55.0±0.4|77.5±0.4|80.0±0.5|**69.7**|
>
> |TerraInc|L100|L38|L43|L46|Avg|
> |--------------|----|---|---|---|----|
> |Adam|42.2±3.4|40.7±1.2|59.9±0.2|35.0±2.8|44.4|
> |SGD|41.8±5.8|39.8±3.9|60.5±2.2|37.5±1.1|44.9|
> |SAM|42.9±3.5|43.0±2.2|60.5±1.6|36.4±1.2|45.7|
> |GENIE(our)|55.2±4.8|47.5±2.1|59.2±0.4|45.9±1.0|**52.0**|
> # A2
> For baseline optimizers, we used results from Zhang et al. (ICCV 2023) obtained under the DomainBed framework. For GENIE, two hyperparameters—dropout probability $p$ and the moving average coefficient $\beta$—were tuned via the standard DomainBed hyperparameter selection procedure (hparams_seed ∈ {0, ..., 19}), selecting configurations with the highest validation accuracy. We further discuss hyperparameter sensitivity in our response to **Reviewer V3wF (A2)**, and will publicly release optimized hyperparameters for reproducibility.
> # A3
> The preconditioning value includes the term  $\tanh(1/\sigma_t^2)$, which tends to suppress the updates of gradients with high variance. While this helps avoid overfitting to noisy gradients, it can undesirably reduce updates to certain parameters. To counterbalance this suppression, we introduce a reciprocal noise scaling factor $1 - \tanh(1/\sigma_t^2)$ which amplifies the noise component where variance is high, encouraging exploration of alternative solutions. This form was chosen to maintain a complementary relationship with the preconditioning factor and encourage escape from sharp or suboptimal minima.
> # A4
> We treated the dropout probability $p$ as a global hyperparameter, applying it uniformly across all layers and parameters for simplicity and consistency. This hyperparameter was tuned individually for each dataset.
>
> ---
> While not explicitly raised as formal questions, we would also like to respectfully address several concerns implied in your review:
> # Feature Invariance
> You correctly noted the lack of feature-level analysis supporting our claim of domain-invariant features. To address this, we have now included t-SNE and UMAP visualizations (fully anonymized): [**URL**](https://imgur.com/a/Pu3xrOi). These results clearly demonstrate improved cross-domain feature alignment and support our claims about domain-invariant feature learning.
> # OSGR Conjecture
> We agree that the conjecture about uniform OSGR leading to better generalization lacked formal proof. However, Section 3.3.1 shows analytically that our proposed preconditioning yields more uniform parameter-wise OSGR, thereby achieving higher total OSGR across all parameters as implied by Jensen's inequality. Additionally, Liu et al. (ICLR 2020) established that higher total OSGR correlates with improved generalization. We recognize that this point is a critical part of our contribution. To further address this concern, we refer to our response to **Reviewer MLrB**, which includes a PAC-Bayesian analysis comparing GENIE with SAM, showing our method explicitly minimizes the KL term for better generalization.
>
> # Sharpness
> We acknowledge your point that we claimed GENIE naturally leads to flatter minima without explicitly measuring curvature. While the SAM method is grounded in PAC-Bayesian theory and focuses on sharpness—which can be interpreted as a bound on empirical risk with respect to the posterior parameter distribution—we would like to highlight that, Our preconditioning strategy provide an additional benefit: it also contributes to tightening the generalization bound by explicitly minimizing the KL divergence term as our answer to **Reviewer MLrB**.
>
> We hope our clarifications adequately address your concerns and demonstrate the rigor and value of our proposed method.

---

> > ### Comment · Reviewer_Z37r · 2025-04-03
> >
> > I thank the authors for all my concerns and recommend accepting the paper for the conference. Please include all additional insights discussed in the rebuttal to the final manuscript.

---

> > > ### Author Response · Authors · 2025-04-05
> > >
> > > Thank you for your positive feedback and constructive comments! We appreciate your recommendation for acceptance, and we will incorporate all the additional insights discussed in the rebuttal into the final manuscript to further improve the work.

---

### Official Review · Reviewer_V3wF · 2025-03-13

**Overall Recommendation:** 4

**Summary:**

This paper proposes a novel optimizer that leverages the one-step generalization ratio to assess each parameter’s contribution to loss reduction, aiming to promote domain-invariant feature learning.

**Claims And Evidence:**

The paper’s claims are clearly stated, and the experiments presented provide convincing evidence in support of these claims.

**Essential References Not Discussed:**

None.

**Experimental Designs Or Analyses:**

The experimental setup is generally solid. However, to further validate the effectiveness of the proposed optimizer, I recommend extending the experiments to segmentation or detection tasks in domain generalization settings (e.g., from GTAV to Cityscapes, BDD100K, and Mapillary). In addition, exploring how different hyperparameters (such as β1 and β2) influence the optimizer’s performance would provide valuable insights into its sensitivity to hyperparameter choices.

**Methods And Evaluation Criteria:**

Both the proposed method and the chosen evaluation metrics align well with the stated research problem.

**Other Comments Or Suggestions:**

None.

**Other Strengths And Weaknesses:**

None.

**Questions For Authors:**

Please address my concerns raised under “Experimental Designs or Analyses.”

**Relation To Broader Scientific Literature:**

The paper references and conducts experiments on domain generalization tasks and includes comparisons with other existing optimizers.

**Theoretical Claims:**

I’ve checked the proofs and no issues are found.

---

> ### Author Rebuttal · Authors · 2025-03-30
>
> Thank you for your valuable comments. Below, we address the points raised under "Experimental Designs or Analyses":
>
> # A1
> Because of limited time and computational resources, we were unable to include experiments on additional tasks such as object detection or segmentation within the submission period. We fully agree that evaluating GENIE on a wider range of tasks (e.g., detection, segmentation, or face anti-spoofing) would further validate its generalization capabilities. We consider this an important direction for future work and plan to release additional results and code for these tasks moving forward.
>
> # A2
> GENIE uses two key hyperparameters: the dropout probability $p$ and a moving average coefficient $\beta$ (used for computing the running mean and variance of gradients). We conducted an additional hyperparameter sensitivity analysis on the OfficeHome dataset to examine the impact of these parameters. As shown by our new experimental results, GENIE consistently outperforms SGD, Adam, and SAM across a range of $p$ and $\beta$ values, demonstrating the method’s robustness to these hyperparameters. (All other training settings were held constant in this analysis.)
>
> We note that we performed a grid search to tune hyperparameters in this new experiment; in contrast, for all other experiments we followed the DomainBed protocol and selected hyperparameters based on validation performance. We hope these clarifications address your concerns. Thank you again for your thoughtful and constructive feedback.
>
> see: [**URL**](https://imgur.com/a/puvgaCx). (The link is fully anonymized.)
>
> ---
> We hope these clarifications address your concerns. Thank you again for your thoughtful and constructive feedback.

---

### Decision · Program_Chairs · 2025-05-01

**Decision:**

Accept (oral)

**Comment:**

The three reviewers acknowledge the importance of this work and recommend acceptance. The reviewers nonetheless raised some questions, most of which were satisfyingly addressed by the authors' feedback. The remaining point to address would be the application of the proposed optimization strategy to other tasks, but the AC acknowledges that, as mentioned by the authors, this may require too much time to do so in the limited timeframe. Altogether, this work is acknowledged as a solid contribution, and the authors are encouraged to incorporate elements of their feedback in the final version.